# Holocene environmental and climate evolution of Central West Patagonia as reconstructed from lacustrine sediments of Meseta Chile Chico (46.5º S, Chile)

Carolina Franco[1], Antonio Maldonado[2,3], Christian Ohlendorf[1], A. Catalina Gebhardt[4], María Eugenia de Porras[5], Amalia Nuevo-Delaunay[6], César Méndez[6], Bernd Zolitschka[1]

[1] University of Bremen, Institute of Geography, GEOPOLAR, Bremen, Germany
[2] Centro de Estudios Avanzados en Zonas Áridas (CEAZA), La Serena, Chile
[3] Departamento de Biología Marina, Universidad Católica del Norte, Coquimbo, Chile
[4] Alfred Wegener Institute Helmholtz Centre for Polar and Marine Research (AWI), Bremerhaven, Germany
[5] IANIGLA, Consejo Nacional de Investigaciones Científicas y Técnicas (CONICET), Mendoza, Argentina
[6] Centro de Investigación en Ecosistemas de la Patagonia (CIEP), Coyhaique, Chile

Correspondence to: Carolina Franco (cafranco@uni-bremen.de)

**Abstract.** Holocene environmental changes in Patagonia were mostly shaped by fluctuating ice-cover recession. Consequently, environmental reconstructions are largely based on discontinuous moraine chronologies from valley deposits. Here, we present a 3 m-long continuous sediment record recovered from Laguna Meseta (LME), a lake located on Meseta Chile Chico. Its altitude and location relative to the North Patagonian Icefield provide a unique opportunity to reconstruct the glacial history and related environmental dynamics.

Our radiocarbon chronology constrains sedimentation to the last ~10,000 years and provides a minimum age for postglacial ice-free lacustrine conditions due to a westward retreat of the ice cap. Lacustrine productivity reached its maximum at the start of the lake phase and decreased afterwards. Between 5,500 and 4,600 cal yr BP, a major shift towards allochthonous sediment accumulation occurred, caused by an abrupt increase in clastic deposition from basaltic lithologies of the Meseta Chile Chico. This episode correlates with the precipitation-driven mid-Holocene glacier advance of Patagonian glaciers and suggests that conditions were colder/wetter on Meseta Chile Chico at that time. After 4,600 cal yr BP, these conditions continued to supply LME with clastic sediments until a stepped decrease around 900 cal yr BP. Thereupon, lacustrine productivity distinctly increased and stabilized around 300 cal yr BP.

Our findings indicate that changes in sedimentation on Meseta Chile Chico were mainly controlled by regional variability of precipitation. Furthermore, strong correlation between our records and available proxies for oscillations of the Southern Hemispheric Westerly Winds suggest a pronounced climatic control by this prominent wind system for Central West Patagonia during the last 10,000 years.

**Keywords: Lake sediments, Central West Patagonia, Middle Holocene, multiproxy analysis, precipitation, tephrochronology.**

## 1 Introduction

The modern landscape of Patagonia is the result of climate-driven fluctuations of the large Patagonian Ice Sheet (Davies et al., 2020) that completely covered the Austral Andes during the Last Glacial Maximum (LGM) (Caldenius, 1932; Hulton et al., 1994; Rabassa et al., 2005). Throughout the Quaternary, glaciers of the Patagonian Ice Sheet advanced and retreated repetitively before splitting into the South and North Patagonian Icefields around 15 ka (Davies et al., 2020; Thorndycraft et al. 2019) as well as into several smaller ice caps distributed along the peaks of the Andean mountain range (Davies et al., 2020; Glasser et al., 2004; Sugden et al., 2005; Turner et al., 2005). However, after the end of accelerated ice recession mostly dated

prior to the Pleistocene-Holocene transition (Bendle et al., 2017a; Davies et al., 2018; García et al., 2019), glacier fluctuations continued to occur.

Holocene climate variability resulted in regional glacier advances, still-stands or recessions, also referred to as "Neoglaciation" (Porter, 2000) that have been mostly studied based on moraine deposits in glacial valleys (Aniya, 1995; Clapperton and Sugden, 1988; Mercer, 1976). However, related chronologies are still subject of debate due to the discontinuity and asynchrony

that entail moraine deposition and erosion as well as local topographic factors influencing glacier dynamics (Glasser et al., 2009).

Conversely, lacustrine sediments are continuous archives of environmental conditions. Therefore, we have chosen to investigate a lacustrine sediment record from Meseta Chile Chico, a basaltic plateau covered by several small lakes. With a multiproxy approach, we aim to 1) constrain sedimentation dynamics of one of these lakes throughout the Holocene; 2)

reconstruct environmental history of the plateau and its surrounding area; and 3) establish correlations among our local environmental changes with regional glacial oscillations, and paleoclimate proxies. The objective is deriving insights into Holocene climatic variability for Central West Patagonia.

**Previous paleoenvironmental reconstructions**

Moraine mapping and chronological studies at the eastern border of lakes General Carrera and Cochrane indicate that Meseta

Chile Chico was located between two major ice lobes of the large Patagonian Ice Sheet, which completely occupied these lake basins during the LGM (Caldenius, 1932; Glasser and Jansson, 2005) between ca. 34-16 ka (Bendle et al., 2017b; Douglass et al., 2005b; Singer et al., 2004; Smedley et al., 2016; Hein et al., 2010). Moreover, ages of post LGM moraines indicate that the General Carrera ice lobe remained at the northern border of the Meseta Chile Chico at least until 14.7 ka (Smedley et al., 2016) before retreating further to the west to develop the General Carrera paleolake basin. Ice front reconstructions from the

opening of the Baker valley surrounding the eastern border of the North Patagonian Icefield (Thorndycraft et al., 2019) point to a separation between the North Patagonian Ice Sheet and the ice cap that covered the Parque Nacional Patagonia around 12.6-10.5 ka (Glasser et al., 2016; Thorndycraft et al., 2019).

Reconstructions of the Patagonian Ice Sheet for Meseta Chile Chico date the retreat from its eastern boundary to between 25-20 ka, but the ice cap that covered the Parque Nacional Patagonia remained active until 10 ka before splitting into the modern

glaciers that occupy the catchment of this range today (Davies et al., 2020). Most of these reconstructions are based on moraine chronologies from the valleys and show a high confidence level for low altitudes, whereas for topographic highs, such as the studied Meseta Chile Chico, these reconstructions are not very precise. Therefore, the timing for deglaciation and consequent lake formation on Meseta Chile Chico still remains unconstrained.

Moraine deposits in the vicinity of the North Patagonian Icefield dated to between 10-6 ka are scarce (Douglass et al., 2005a;

Harrison et al., 2017; Sagredo et al., 2021). They indicate that during this period the Patagonian landscape was mainly formed by ice recession and splitting-up of this major ice sheet into several smaller glaciers covering regional topographic highs. Consequently, and as documented by several pollen records, large *Nothofagus* forests developed around Lago Cochrane supporting the existence of higher temperatures and/or higher effective moisture for this period (Villa-Martínez et al., 2012; Maldonado et al., 2022; Henríquez et al., 2017; Iglesias et al., 2018).

During the Middle Holocene, climate conditions in Central West Patagonia are regionally recorded as a period of increased precipitation and reduced seasonality linked to a persistent yearlong influence of the Southern Hemisphere Westerly Winds (SWW, Markgraf et al., 2007; de Porras et al., 2012; de Porras et al., 2014; Nanavati et al., 2019; Villa-Martínez and Moreno, 2021; Maldonado et al., 2022). These climatic conditions were responsible for glacial advances between 6-4 ka throughout Patagonia (Davies et al., 2020; Mercer, 1976; Glasser et al., 2004; Aniya, 1995; Kaplan et al., 2016). Moraine deposits from

this period are found around the North Patagonian Icefield (Fernández et al., 2012; Harrison et al., 2012; Nimick et al., 2016; Bertrand et al., 2012) and Cerro San Lorenzo in Central Patagonia (Sagredo et al., 2021; Sagredo et al., 2018), around the

Southern Patagonian Icefield (Reynhout et al., 2019; Strelin et al., 2014; Kaplan et al., 2016), as well as in the Darwin Cordillera in South Patagonia (Reynhout et al., 2021) and on the northern Antarctic Peninsula (Kaplan et al., 2020). This stage could have also triggered a series of large dam-breaching events in several lakes of the Rio Baker catchment, as documented by dated glacial lake outburst flood (GLOF) deposits (Benito et al., 2021).

Transitioning to the Late Holocene, paleoenvironmental patterns in Central West Patagonia likely resulted from an enhanced summer insolation compared to present levels (Alder and Hostetler, 2015; Nanavati et al., 2019) superimposed on an intensified seasonal and/or interannual precipitation variability. These conditions are indicative of highly variable SWW presumably related to an increased frequency at annual to secular timescales, like they are reported for the Southern Annular Mode (SAM) at local (Moreno et al., 2018; Moreno et al., 2014) and regional scales (Abram et al., 2014; Dätwyler et al., 2018).

For this period, moraine chronologies as well as historical records of glacier advances have been recognized from North to South Patagonia (Mercer, 1976; Glasser et al., 2005; Garibotti and Villalba, 2009; Davies and Glasser, 2012; Kaplan et al., 2016). This suggests that the latest neoglacial stage occurred at ca. 0.5-0.2 ka (Davies et al., 2020), overlapping with the Northern Hemisphere "Little Ice Age" (Grove, 2004). In opposition with records from the Northern Hemisphere, where this period is documented as the strongest Holocene glacial advance , geomorphological evidences from Patagonia indicate that it was of minor intensity in comparison with the Middle Holocene glacial advance (Glasser et al., 2005; Kaplan et al., 2016).

Although several studies on deglaciation and neoglaciation have been carried out at the borders of the North Patagonian Icefield (as well as in northern and southern areas of Patagonia) and in low altitudes such as regional valleys and lakes, the Meseta Chile Chico and its surrounding ranges have not yet been considered for paleoenvironmental studies.

## 2 Regional Setting

### 2.1 Geomorphology and modern setting

The Meseta Chile Chico is located in Central West Patagonia in the Región de Aysén, Chile (Fig. 1a), ca. 100 km east of the eastern margin of the North Patagonian Icefield (Fig. 1b). It is placed on the eastward slope of the easternmost mountain range of the Andes, the Parque Nacional Patagonia Range (Fig. 1c). Positioned between 1,100 and 2,000 m a.s.l., the plateau extends from Cerro Pico Sur in the north to the western slopes of the valley of Río Jeinemeni.

Due to its altitude and morphology, the environment of Meseta Chile Chico is less influenced by local factors such as surrounding topography, vegetation and human intervention (Méndez et al., 2023) compared to the surrounding glacial valleys. Additionally, this plateau is covered with several small lakes (Fig. 2a), interconnected by meadows and ephemeral rivers. These lakes are part of its drainage basin, which currently drains into Río Jeinemeni. Today, this basin it is mostly fed by snowmelt and rain. Nevertheless, several (today empty) glacial cirques that occur in the catchment of Meseta Chile Chico as well as their topographic connections with the Cerro Las Nieves glaciers to the west, indicate past glacial influence for this area (Fig. 1c).

Our study focuses on one of the lakes located on the Meseta Chile Chico, unofficially referred to as Laguna Meseta (LME: 46.716º S, 71.844º W, 1,457 m a.s.l.). LME covers a surface area of ~22,600 m$^2$ and reaches a maximum water depth of ~7 m (Fig. 2b). It comprises a catchment area of  ~675,000 m$^2$ and belongs to the Arroyo Cardenio fluvial basin, which constitutes the major river channel of the plateau.

The vegetation surrounding LME is a grass-shrub steppe dominated by *Festuca pallescens* and *Azorella prolifera* associated with *Acaena splendens, Adesmia boronioides, Baccharis patagonica*. Dwarf shrubs such as *Nassauvia dentata, Senecio triodon, S. poeppigii, Empetrum rubrum, Perezia pericularidifolia* are also important elements given the proximity of LME to the ecotone with high Andean vegetation (Luebert and Pliscoff, 2017).

## 2.2 Geology

Most of Meseta Chile Chico is formed by alkaline flood basalts of Miocene age, defined as the Upper Basalts of Meseta Chile Chico. Furthermore, at the northern border of Meseta Chile Chico, in the watershed of Arroyo Cardenio and in the Cerro las Nieves-Cerro Pico Sur range as well as at some of its eastern flanks, older Paleocene-Eocene basaltic rocks of the Lower Basalts of the Meseta Chile Chico crop out (Fig. 1c, Espinoza et al., 2005). These rocks are (partly) covered by unconsolidated Holocene mass movement deposits, described as disordered and fragmented sediments, which range from metric blocks to clay grainsizes (De La Cruz y Suárez, 2008). Their distribution along the entire Meseta Chile Chico region, following the topographic slopes, is likely influenced by postglacial stress release among other factors (Pánek et al., 2021).

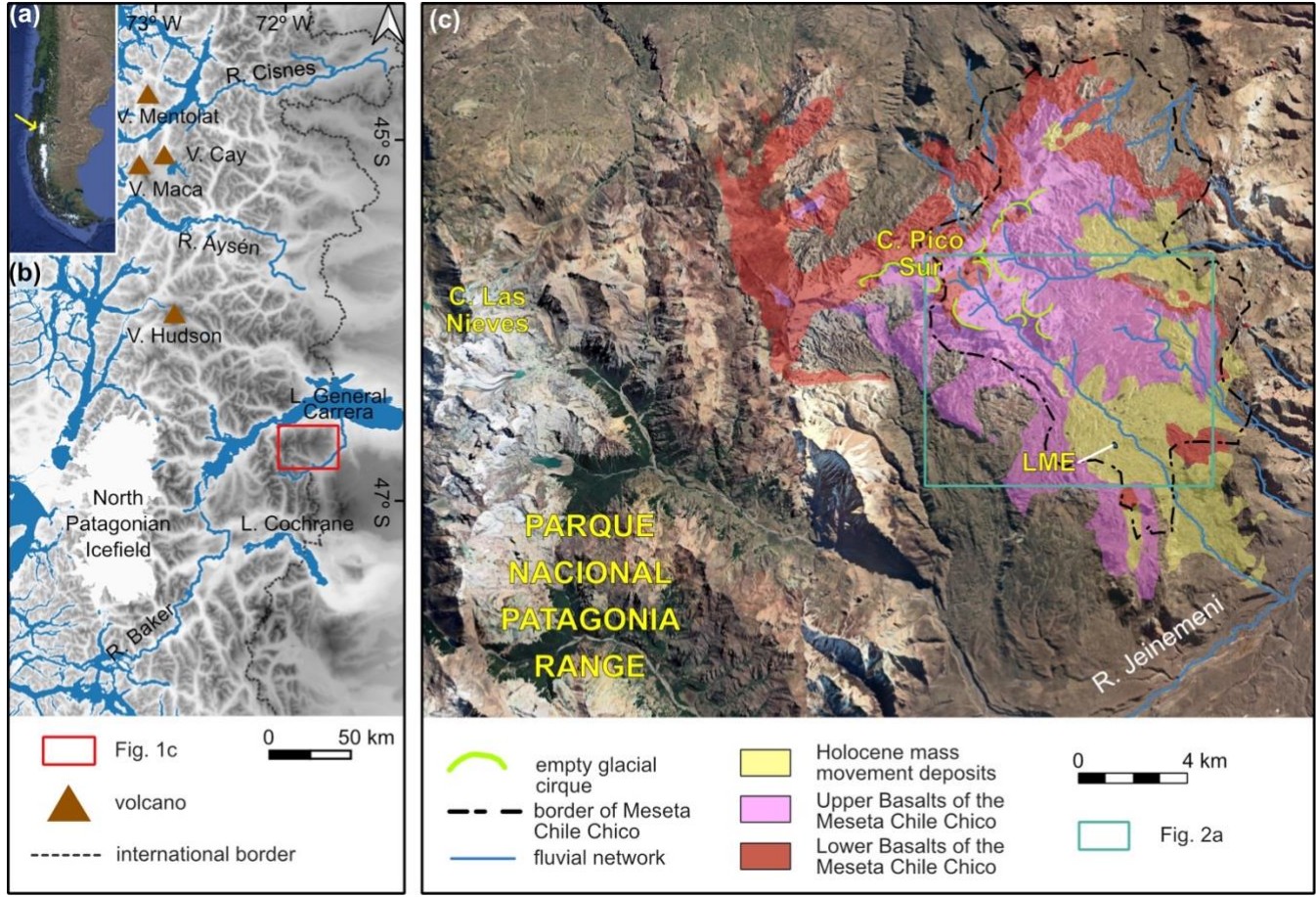

Figure 1 (a) Location of the North (yellow arrow) and South Patagonian Icefields in South America. Background corresponds to the Global Multi-resolution Terrain Elevation Data (GMTED 2010); (b) Location of the study site in Central Patagonia with location of volcanoes (V.) and lakes (L.) mentioned in the text. Background image corresponds to © Google Satellite (2023); (c) Main geological (De La Cruz and Suárez, 2008) and geomorphological units of Meseta Chile Chico (this work) with location of Laguna Meseta (LME). Other abbreviations: R.: río (river); C.: cerro (peak).

Another relevant geological feature of Chile Chico Meseta is its proximity to volcanoes of the Southern Volcanic Zone (Futa and Stern, 1988), such as Hudson, Macá, Cay and Mentolat volcanoes (Fig. 1b). Several Holocene eruptions have been documented between Río Cisnes and Lago Cochrane in surface deposits as well as in lake and fjord sediments (Stern et al., 2016; Weller et al., 2017). The most relevant of these events corresponds to the eruption of Hudson (H1) dated to between 8,530-7,750 cal yr BP. Most likely, this was the largest Holocene eruption for Central Patagonia (Naranjo and Stern, 2004).

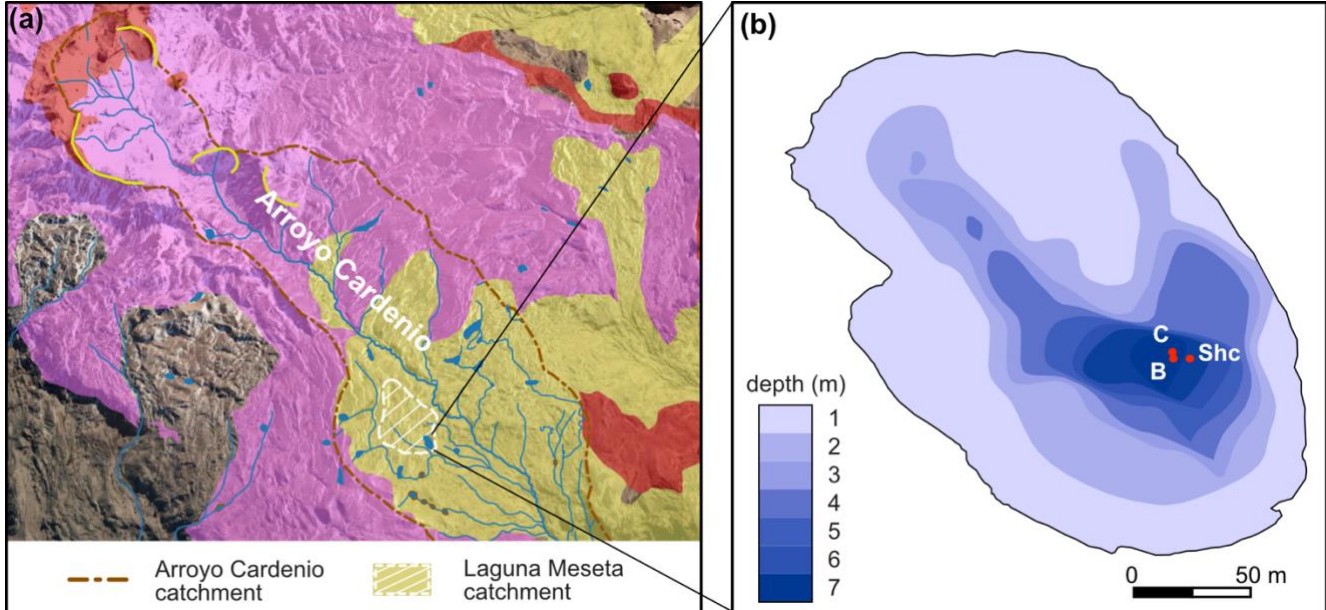

Figure 2 (a) Catchments of Arroyo Cardenio and Laguna Meseta (for location see blue frame of Fig. 1c). For colour signatures refer to Fig. 1c. Background image corresponds to © Google Satellite (2023); (b) Bathymetry of Laguna Meseta and location of sediment cores LME–Shc, LME-B and LME–C.

## 3 Methods and Materials

### 3.1 Bathymetry, sediment coring and subsampling

In March 2020, bathymetric analysis was performed at Laguna Meseta with a GPSmap 521s GARMIN echo sounder along different transects that followed a gridded pattern across the lake. Around 300 data points, including geographic coordinates and water depth, were recorded and interpolated, and plotted with QGIS in order to select the best point for retrieving sediment cores.

Following bathymetric measurements, three sediment cores with in total eight sections were collected at LME from the maximum water depth of 7 m at adjacent coring series (Fig. 2b). Cores LME-B and LME-C were extracted with a modified Livingston piston corer (diameter: 5 cm), whereas core LME-Shc was recovered with a UWITEC gravity corer (diameter 9.6 cm).

The cores were cut along their depth axis by the lab team of the Center of Advanced Studies in Arid Zones - CEAZA (La Serena, Chile), obtaining two identical halves for each core section. One of the halves remained at CEAZA for ongoing pollen analyses and the other was sent to the GEOPOLAR lab (University of Bremen, Germany). In Bremen, core halves were described considering colour (applying the Munsell Soil Colour Charts), lithology, grainsize and pyroclastic fall-out deposits. After conservative (i.e., non-destructive) techniques such as downcore X-ray fluorescence (XRF) scanning, X-ray radiography as well as magnetic susceptibility logging were performed, a 3 m-long composite sediment record was constructed with the additional help of six visible tephra layers as marker horizons (the Supplement, Fig. S1). Subsampling for additional parameters was carried out with 1-10 cm spatial resolution. All samples were immediately freeze-dried for further processing.

### 3.2 XRF core scanning

Downcore X-ray fluorescence (XRF) scanning was performed on untreated core halves using the ITRAX XRF core-scanner (Croudace et al., 2006) at the GEOPOLAR lab (University of Bremen) with a Mo-tube that was run at 30 kV and 45 mA. A step size of 1 mm and a dwell time of 5 s per step were applied. This ensured average total count rates of approximately 50 kcps. To assure optimal data quality, a mathematical model (setting file) was fitted to a sum spectrum that contains all spectra

recorded. Fitting was done with the ITRAX proprietary software Q-Spec (version 8.6.0). Element integrals (peak areas) were determined during batch processing using the designated setting file.

After assessing and cleaning the elemental data for outliers, five elements were selected for interpretation: Si, K, Ca, Ti and Sr (the Supplement, Fig. S2). For these elements a centred log-ratio (clr) transformation (Weltje et al., 2015) was applied to avoid effects of intensity variability that may be caused by mutual dilution effects of the different elements present in the sediment core. Downcore profiles are displayed as clr transformed element curves or as logarithmic elemental ratios throughout this publication. Running means were calculated with a moving window of 10 data points and for data gaps a linear interpolation was used instead. For organic matter (OM) content estimations, the logarithmic XRF incoherent/coherent ratio was (Guyard et al., 2007; Davies et al., 2015) used after ground-truthing with $LOI_{550}$ data for a subset of samples.

### 3.3 Magnetic susceptibility

Magnetic susceptibility was measured at the Alfred Wegener Institute Helmholtz Centre for Polar and Marine Research in Bremerhaven, Germany. Measurements were carried out on half cores with 1 cm resolution on a Geotek Multi-Sensor Core Logger (Geotek Ltd, UK) using a Bartington MS2E sensor.

### 3.4 Geochemistry of tephra layers

Six tephra layers (labelled T1-T6 from bottom to top) were identified on the basis of macroscopic and microscopic (smear slides) sediment description as well as using radiographs and XRF core-scanning data. Due to their compositional variability, no element ratio correlates with all six tephra layers. Furthermore, data points corresponding to these layers were discarded for environmental interpretation.

From the basal section of the thickest (14.5 cm) tephra deposit, two bulk samples enriched in pumice were obtained. After freeze-drying, the samples were pulverized and fused with lithium tetraborate to produce glass beads that were analysed for major and trace elements with wavelength dispersive X-ray fluorescence spectroscopy (Panalytical Axios[max]) at the University of Oldenburg. A Total Alkali Silica (TAS) diagram for tephra chemistry was plotted with the TAS template for Excel 1.1 (Stosch (2022).

### 3.5 Radiocarbon dating and age-depth modelling

A total of eight bulk sediment samples were obtained (Table 2) at the CEAZA lab and sent to the Direct AMS laboratory (Washington State, US) for radiocarbon dating.

An age-depth model was obtained with rbacon (Blaauw and Christen, 2011) version 2.5.7 using the Southern Hemisphere calibration curve SHCal20 (Hogg et al., 2020). Deposits with tephra layers were modelled as slumps, as they represent episodes of instantaneous (event) deposition.

### 3.6 Bulk geochemistry

Loss on ignition (LOI) at 550 ℃ was measured at the CEAZA laboratory with 1 cm intervals to determine OM. Additionally, total carbon (TC) was measured every 5 cm, whereas total inorganic (TIC) and total organic (TOC) carbon percentages were measured every 10 cm with a CNS elemental analyser (EuroEA) at the GEOPOLAR lab (University of Bremen). Moreover, biogenic silica (BiSi) was measured in 2 cm steps at the GEOPOLAR lab using the automated leaching method of (Müller and Schneider, 1993).

**3.7 Grainsize**

Grainsize distributions were measured in 2 cm resolution (excluding tephra and other layers with high content of pyroclastic components) with a Beckman Coulter LS 200 laser granulometer. All samples were dried and homogenized prior to analysis. About 0.7 g of each sample was weighed into 50 ml centrifuge tubes. During pre-treatment, organic components were removed with a 30 % hydrogen peroxide solution at 105 °C. Biogenic opal (diatoms) was dissolved with a 2 mol/l sodium hydroxide solution at 85 °C. After being washed and centrifuged, the samples were shaken over night with a dispersion agent on a Na-polyphosphate base to ensure a proper dispersion of the grains. Each sample was treated with a sonication probe for 30 s directly before measurement and then analysed for 60 s several times until three comparable replicates have been obtained.

**3.8 Compiled chronologies**

Published cosmogenic and optically stimulated luminescence (OSL) ages cited throughout this work are presented as their means in ka as calculated by Davies et al. (2020) after being reviewed in their original publication. Radiocarbon ages published prior to the availability of the SHcal20 calibration curve (Hogg et al., 2020) were recalibrated. Cosmogenic and OSL chronologies, as well as time periods defined by a mix of different dating techniques, are always presented as "ka".

**4 Results**

**4.1 Lithology**

By means of macroscopic description as well as magnetic susceptibility (MS) and grainsize analysis, the lithology of the Laguna Meseta composite profile (LME-CP) was grouped into nine lithological units (Units A-H and T5), of which two (F, T5) are dominated by tephra (Fig. 3).

For the whole composite profile, the biogenic content is characterized by unidentifiable plant remains, lenses of OM and diatoms, which are significantly more abundant within silty lithological units H to D. Similarly, glass shards are observed in all lithological units but particularly abundant in units G and B. Glass, pumice and obsidian fragments between 0.5 and 1-mm size as well as fragmented crystals of pyroclastic origin are found throughout and mixed with the sediment.

The lowermost sediments of LME-CP are very dark greyish brown silts that form unit H (292.5 to 240.5 cm composite depth). This unit is mainly composed of laminated organic (diatomaceous) sediments with abundant traces of plant remains up to a few millimeters in size and dark grey lenses of OM.

Unit G (240.5-229 cm composite depth) is made of dark greyish brown, clayey to silty very fine sand, containing plant remains, as well as glass and crystalline fragments of quartz and feldspars. The boundary between sedimentological units H and G is notably sharp, yet we do not interpret this as erosional. Rather, we attribute this change in sediment properties to the abundance of fragmented materials, which indicate a pyroclastic origin related to the remobilization of T6. Despite this, lacustrine sediments remain the primary component of this core segment. Hence, for the purpose of this study, it is considered as a lithological unit.

Unit T5 (229-214.5 cm composite depth) corresponds to a 14.5 cm thick pyroclastic layer and therefore is defined as a separate lithological unit, but not considered for environmental interpretation. Because of thicknesses <1.7 cm, the remaining tephra layers (T1-T4 and T6) are not defined as lithological units. The tephra record is addressed in more detail in the following subsection.

Between 214.5 and 199.6 cm composite depth, Unit F lies immediately on top of T5. This unit corresponds to a distinctly laminated section, formed by layers of vitric ash that alternate with mixed layers of ash and clay. Because the major component of this section is pyroclastic material, together with the deposit assigned to T5, Unit F is also excluded from the age-depth model.

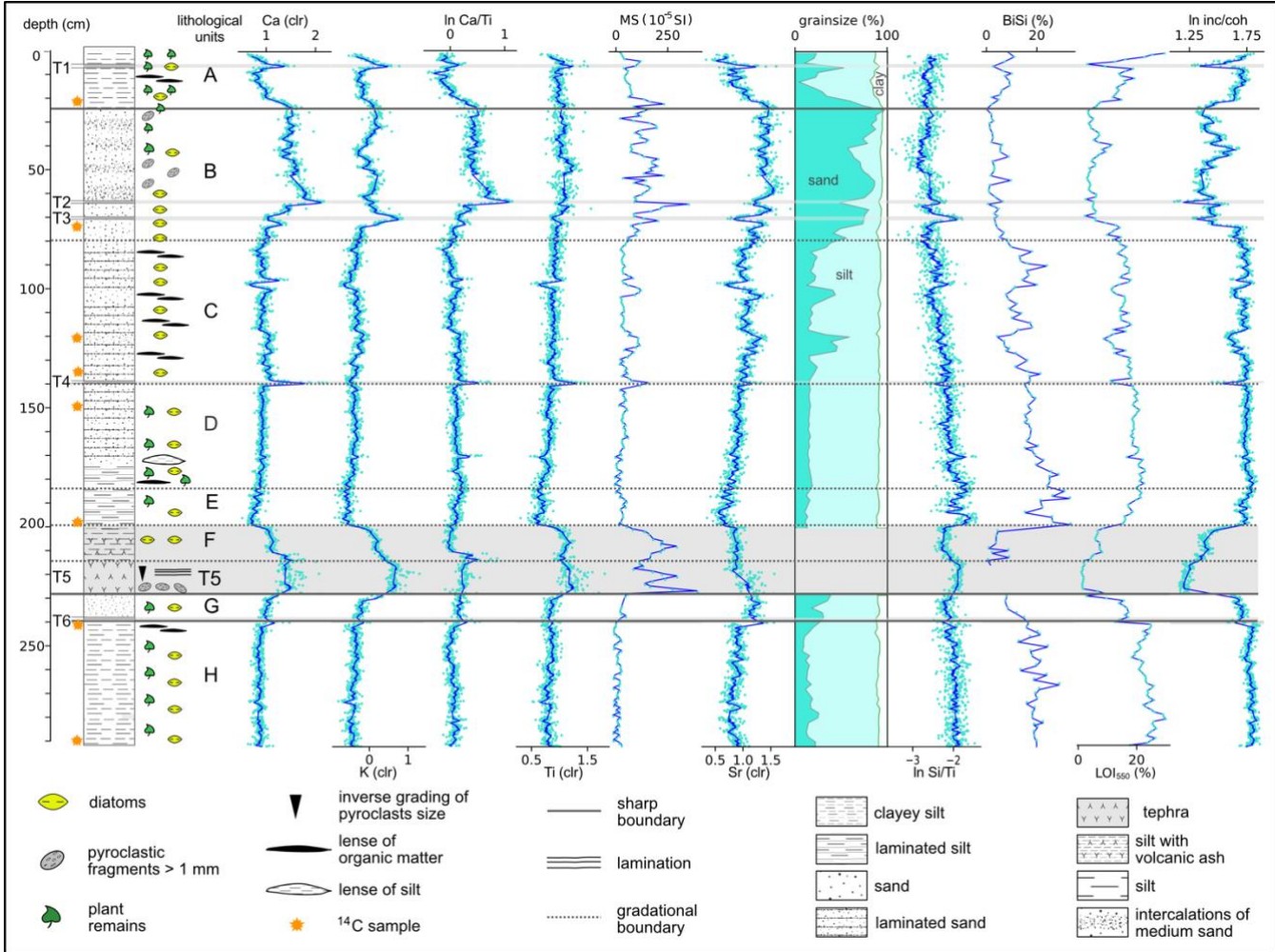

Figure 3 Lithology, sedimentary structures, lithological units, and main elemental components (clr values and logarithmic ratios) with data points (light blue dots) and running means (dark blue lines), magnetic susceptibility (MS), grainsize, biogenic silica (BiSi), loss on ignition at 550 ° C (LOI$_{550}$) and the logarithmic incoherent-coherent ratio (inc/coh) of the composite sediment record vs depth.

Unit E (199.6-184 cm composite depth) includes very dark to dark greyish brown silt beds with intense lamination of dark grey and brown layers (1 to 3 mm-thick). Lamination becomes less pervasive and thinner towards the top. Plant remains and organic lenses as well as lenticular beds of light brown silt are recognized.

Unit D is formed by very dark brown, laminated sandy silt to silt and ranges from 184 to 139.7 cm composite depth. This unit also contains high amounts of organic components, but unlike Unit C does not present intercalations of medium sand.

Unit C is enriched in organic components, plant remains and dark brown lenses of OM. It ranges from 139.7 to 78.6 cm composite depth and is a transitional unit between units D and B. It is formed by very dark brown, diffusely laminated sandy silt to clayey silt with irregularly intercalated beds of sublithic fine sand.

Unit B (78.6-24.1 cm composite depth) is composed of very dark greyish brown, fine to very fine sublithic sand, with intercalated 0.5 to 2 cm-thick beds of medium to coarse, subarcosic sand. Its composition is highly minerogenic with subangular quartz and feldspar as main components, and volcanic lithics as secondary components. Micaceous minerals, felsic volcanic clasts, unrecognized lithic clasts, and plant remains are subordinated. Some lenses of very fine sand are observed. Near to its top, grainsize decreases progressively and the transition to unit A is gradual.

Finally, Unit A spans the upper 24.1 cm of the profile and is composed of dark yellowish brown sandy to clayey silt with millimetric plant remains. Lenses of light brown clay (1 cm thick) and dark brown lenses of OM are also observed.

## 4.2 Tephra records

The profile LME-CP contains six tephra layers (T6 to T1, Fig. 3). T5 is most prominent with a thickness of 14.5 cm, whereas the other five tephra layers have thicknesses that range between 8 and 17 mm. XRF scanning results show a non-uniform composition of these layers: T2 and T4 are enriched in Ca and poor in K, T3 shows the opposite trend, whereas T1, T5 and T6 are enriched in Ca and K, while Ti is enriched only for T1, T4 and T5 (Fig. 3).

Within LME-CP, T5 stands out for its thickness and a laminated vitric lapilli deposit at the base, which grades into ash-sized pyroclastic fragments and elongated brownish glass shards. It shows a sharp lower boundary, meanwhile its upper section exhibits distinct laminations marked by alternating coarse and fine ash layers, gradually transitioning into the sediments of Unit E. Moreover, this fall-out deposit is highly enriched in coarse pumice fragments (~2 mm) considered representative of the geochemical composition of its parental magma. These characteristics provide a unique opportunity for our tephra record to discuss a correlation of T5 with regional eruptive events.

According to our age-depth model (Subchapter 4.3), the age of T5 ranges between 8,987 and 8,018 cal yr BP with a mean age of 8,277 cal yr BP. Other tephrochronological studies of this time interval show Holocene eruptive activity of four volcanoes for Central West Patagonia, corresponding from north to south to Mentolat, Cay, Macá and Hudson volcanoes. However, southernmost deposits of Macá and Cay volcanoes have been described in the upper Río Aysén valley, located approximately 130 km North of Laguna Meseta (Fig. 1b, Weller et al., 2017), suggesting that these deposits might have not extended further southward to Meseta Chile Chico. Moreover, the only large eruptive event recognized for Macá volcano corresponds to a Late Holocene event (Weller et al., 2017; Naranjo and Stern, 2004), which is too young to be correlated with T5. For Cay volcano no large regional events are documented. It is suggested that Cay volcano had no important activity during the Holocene at all (Weller et al., 2019; Weller et al., 2015).

To the contrary, volcanoes Mentolat and Hudson present large magnitude eruptive events for the Early Holocene (Naranjo and Stern, 1998, 2004), indicating a likely correlation with this study. For the Hudson 1 (H1) eruption an age range between 8,585 and 8,200 cal yr BP (mean age of 8,415 cal yr BP, Stern et al. 2016) was previously documented for the catchment of Lago Cochrane (Maldonado et al., 2022; Naranjo and Stern, 1998, 2004; Stern et al., 2016) and the Rio Zeballos valley (McCulloch et al., 2017), while studies located between Lago Cochrane and Río Cisnes date the Mentolat 1 eruption (Men1) to between 7,864 and 7,399 cal yr BP (Stern et al., 2016; Naranjo and Stern, 2004).

Table 1 Major element composition (all values as weight %) for Hudson H1 bulk samples from the Lago Cochrane area (Stern et al., 2016) and from Meseta Chile Chico (this work).

| Lago Cochrane area | | | | | | | | | | |
|---|---|---|---|---|---|---|---|---|---|---|
| sample name | $SiO_2$ | $TiO_2$ | $Al_2O_3$ | $Fe_2O_3$ | MnO | MgO | CaO | $Na_2O$ | $K_2O$ | $P_2O_5$ |
| 94t-44 | 62.15 | 1.41 | 16.24 | 4.97 | 0.16 | 1.69 | 3.65 | 5.60 | 2.57 | 0.32 |
| | | | | | | | | | | |
| **Meseta Chile Chico** | | | | | | | | | | |
| CT3 120-145 | 55.57 | 1.40 | 15.80 | 7.03 | 0.16 | 2.63 | 4.87 | 4.74 | 1.74 | 0.48 |
| CT3 95-120 | 56.90 | 1.44 | 15.33 | 7.21 | 0.17 | 2.56 | 4.40 | 4.80 | 1.92 | 0.51 |
| average values (this work) | 56.24 | 1.42 | 15.57 | 7.12 | 0.16 | 2.60 | 4.64 | 4.77 | 1.83 | 0.49 |

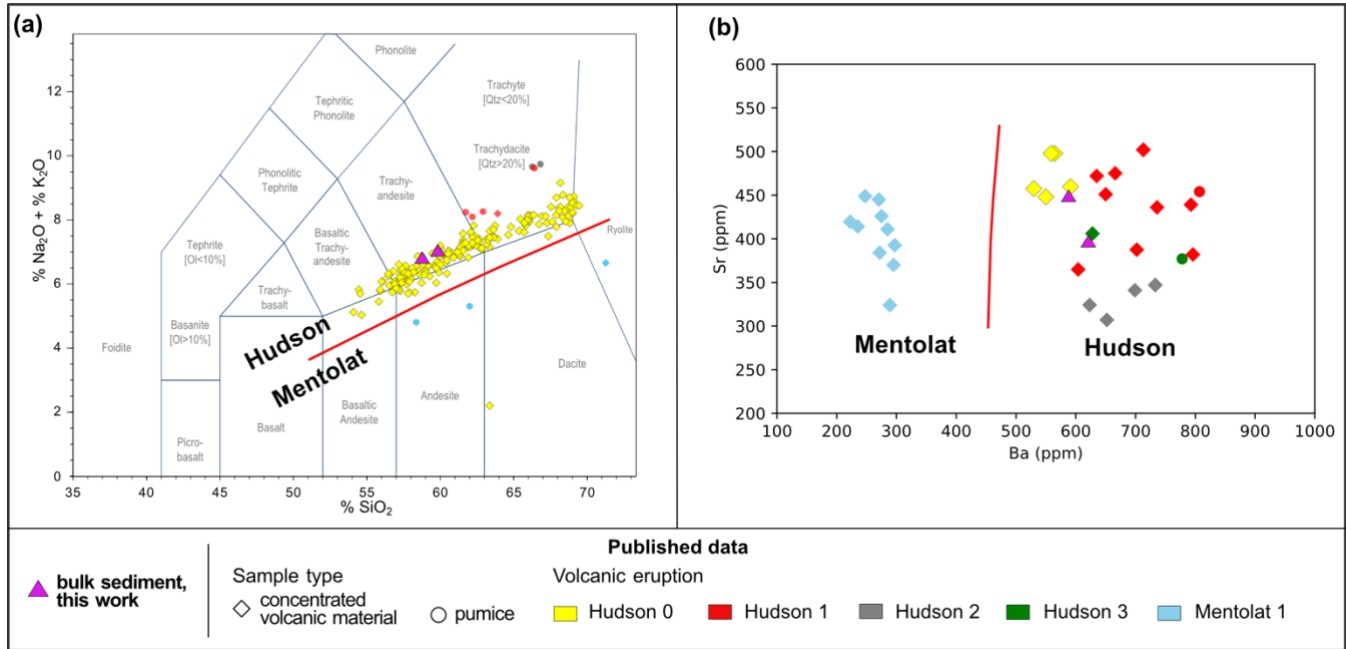

Figure 4 Geochemical characterization of tephra from Hudson and Mentolat volcanoes. (a) Total Alkali Silica (TAS) volcanic classification diagram (Bas et al., 1986) and b) Sr vs Ba concentrations for tephra layers identified around Lago General Carrera and Lago Cochrane. (Hudson 0: Bendle et al., 2017a; Hudson 1: Naranjo and Stern, 1998, Stern et al., 2016; Mentolat 1: Naranjo and Stern, 2004, Stern et al., 2016; Hudson 2, 3: Stern et al., 2016).

Geochemical analysis performed in this study (Table 1) indicate that T5 has a trachy-andesitic composition (Fig. 4a). To address the volcanic provenance of this tephra, we compared its chemical composition with all available records from the volcanic products of both Hudson and Mentolat volcanoes. Our results indicate that Unit T5 falls within the field of alkali-enriched magmatism (Fig. 4a) that distinguishes the Hudson volcano from other eruptive centres of the Southern Volcanic Zone such as Mentolat volcano (Futa and Stern, 1988; Lopez-Escobar et al., 1993; Stern, 1991). Moreover, trace element concentrations are characterized by a high Ba/Sr ratio (Fig. 4b) consistent with pyroclastic deposits (Bendle et al., 2017a; Naranjo and Stern, 1998, 2004; Stern et al., 2016) and volcanic rocks from the Hudson volcano (Stern, 1991, 2008). In contrast, Ba concentrations for the products of Mentolat are significantly lower (Naranjo and Stern, 2004; Stern et al., 2016).

Our results indicate that the chronology as well as the geochemical characterisations of Unit T5 agree with data from the H1 eruptional event. The slightly lower elemental percentages of our samples in comparison to published concentrations for H1 (Fig. 4a) are likely influenced by sample treatment. Chemical compositions from previous studies derive directly from pumice fragments collected in surface pyroclastic deposits (Naranjo and Stern, 2004, 1998) or from sediment samples, from which volcanic material was separated (Stern et al., 2016; Bendle et al., 2017a). Since no removal of the sediment matrix was performed for this study before the analysis, our samples are "less enriched" in $SiO_2$, $K_2O$ and $Na_2O$ compared to the literature. After all these considerations, we assign the T5 tephra to the H1 eruption of the Hudson Volcano.

### 4.3 Chronology

Eight radiocarbon dates were obtained and applied for age-depth modelling (Table 2, Fig. 5). The mean calibrated ages range between 621 and 9,921 cal yr BP and consistently increase with depth. Therefore, all ages were considered for calculation of the age-depth model.

Table 2  AMS radiocarbon dates from the sediment record of Laguna Meseta.

| Core section | Section depth (cm) | Composite depth (cm) | Sample code | Sample type | Radiocarbon age ($^{14}$C BP± 1σ) | Calibrated mean (cal yr BP) | 2σ range calibrated ages (cal yr BP) |
|---|---|---|---|---|---|---|---|
| Shc | 20.5 | 20.5 | D-AMS 043184 | sediment (bulk) | 730 ± 18 | 621 | 670-566 |
| Shc | 71 | 71 | D-AMS 043185 | sediment (bulk) | 4,555 ± 27 | 5,168 | 5,311-5,047 |
| Shc | 117 | 117 | D-AMS 043186 | sediment (bulk) | 5,737 ± 26 | 6,487 | 6,400-6,622 |
| CT2 | 86 | 130.4 | D-AMS 043187 | sediment (bulk) | 6,208 ± 26 | 7,069 | 7,164-6,955 |
| CT3 | 8.5 | 141 | D-AMS 043188 | sediment (bulk) | 6,499 ± 29 | 7,368 | 7,430-7,280 |
| CT3 | 65 | 198 | D-AMS 043189 | sediment (bulk) | 7,261 ± 42 | 8,049 | 8,171-7,958 |
| CT4 | 21.5 | 240.5 | D-AMS 043190 | sediment (bulk) | 8,579 ± 35 | 9,517 | 9,550-9,472 |
| CT4 | 71.5 | 290 | D-AMS 043191 | sediment (bulk) | 8,869 ± 32 | 9,921 | 10,150-9,702 |

Furthermore, T5 assigned to the Hudson H1 eruption, serves as a temporal control point for the $^{14}$C-based age-depth model. The 14.5 cm of sediment above T5 (Unit F, Fig. 3) are interpreted as result of remobilization of H1 fall-out in the lake catchment. This layer is considered contemporaneous with the eruption and, therefore, modelled together with T5 as one instantaneous event. However, it remains unclear whether Unit F represents an event-like deposit or sedimentation over a significant time period. Additionally, the lack of a sharp contact of the upper boundary of Unit F makes it undeterminable at which depth "normal" lake sedimentation resumed.

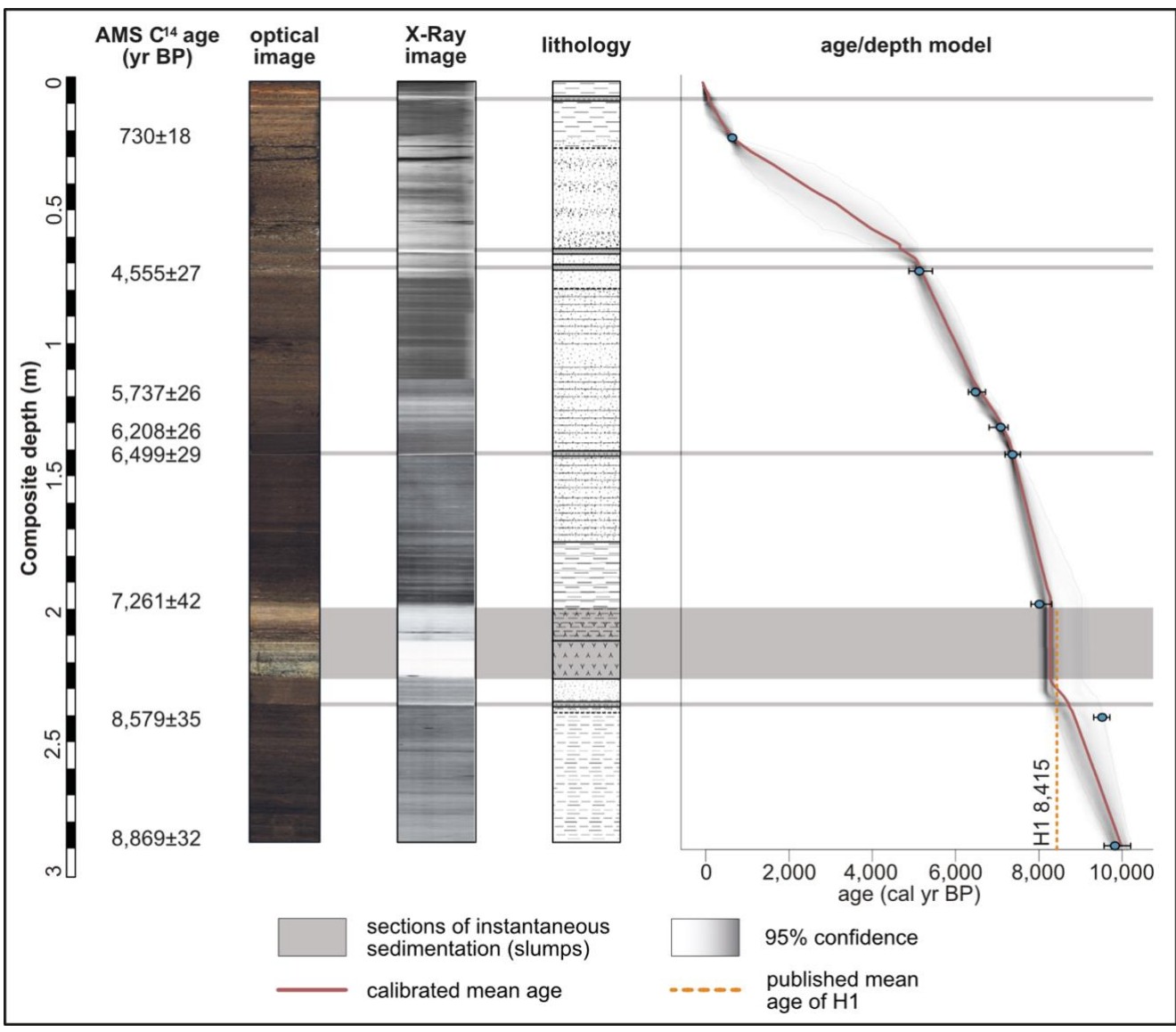

Figure 5 Radiocarbon ages, optical photography, radiography, lithology, and age-depth model of the composite sediment record for Laguna Meseta with the mean age for the Hudson H1 tephra (Stern et al., 2016). The age-depth model was developed

with the rbacon software 2.5.7 applying the southern hemispheric calibration curve SHCal20 (Hogg et al., 2020). Symbols for lithological patterns are displayed with Figure 3.

Published chronologies for H1 indicate a mean age of 8,415 cal yr BP with an age range of from 8,585 to 8,200 cal yr BP (Stern et al., 2016). We consider that this age range estimate, along with the temporal uncertainties associated with Unit F, are not compatible with age-depth modelling. Therefore, the published H1 age was not included in our age-depth model. Nonetheless, our radiocarbon chronology, which provides an age range between 8,987 and 8,018 cal yr BP for T5, is in agreement with the timing of Hudson H1 according to tephrochronological studies.

The radiocarbon age of 8,721 cal yr BP from 240.5 cm composite depth (D-AMS 043190, Table 2) slightly deviates from the calculated mean of our age-depth model. We consider that this offset does not contradict our chronology and might be related to older previously remobilized (by bioturbation?) material within the bulk sample. Another possible explanation could be that our age model overestimates the duration of the H1-Unit F transition (see discussion above). A shorter event would allow a smoother transition between ages at 198 (D-AMS 043189) and 240.5 cm composite depth, therefore its deviation from the

mean values would be smaller. Nevertheless, as this age falls within the 95% confidence interval of our age-depth model it is considered as reliable.

**4.4 Siliciclastic elemental composition, grainsize and magnetic susceptibility**

The highest mean values of the selected element K correspond to tephras T1, T3 and T5. Meanwhile, element Ti reaches its largest peaks in T1, T3 and T5 tephra deposits. These values for K and Ti are related to a high content in volcanic glass and/or

crystals (Fig. 3).

Units H, G, E and D consistently show very low values of elemental concentrations. Consequently, they exhibit more OM represented by the inc/coh ratio. In the case of Unit G, the slightly higher values for Ca, Ti and K are most likely associated to a higher content of pyroclastic crystalline components (reworked T6). Therefore, this increase is not associated to environmental influences.

Throughout Unit C, concentrations of elements progressively increase until reaching the base of Unit B. Although the chemical composition at the base of Unit B is obscured by T3 and T4, it is noticeable that values for all considered elements peak within the lithological units of our records. The chemical content remains high in the sediments above the tephra layers, while the inc/coh ratio reaches a minimum.

In Unit A elemental concentrations drop once again to minimum values (except for T1) with a stepwise increase in inc/coh.

As expected, MS covaries with elemental concentrations throughout LME-CP. Its maximum values are determined for tephra layers (T1-T6) and Unit F. Grainsize measurements, which were not performed for the tephra layers and Unit F, document maximum sand values for the highly minerogenic Unit B (Fig. 3). Minimum values for MS and high fine-grained particles percentages are determined at the base of the core. Unit C increases in sand content, which correlates well with MS, and both parameters increase until Unit B and drop synchronously at the contact between Units B and A.

**4.5 Organic Proxies**

TIC concentrations, considered as estimation of carbonates, are below 1 % for all measured samples. This aligns with our microscopic observation, as no calcite crystals or any other carbonate minerals or carbonaceous organic remains were identified. Hence, the presence of carbonates in the sediments can be considered negligible. Consequently, TOC and TC percentages are regarded as identical confirming that the carbon content of the samples originates primarily from organic

remains. $LOI_{550}$ analyses show a high degree of agreement ($R^2 = 0.92$, Fig. 6a) with TC ≈ TOC values, reflecting that $LOI_{550}$ results are a reliable indicator for the presence of organic matter in the sediments.

The high correlation of the inc/coh ratio with LOI$_{550}$ (R$^2$ = 0.87, Fig. 6b), indicates that inc/coh ratios derived from XRF core-scanning with 1 mm spatial resolution can be used as a proxy for OM as has been shown for other lacustrine sediments (Davies et al., 2015; Guyard et al., 2007; Burnett et al., 2011).

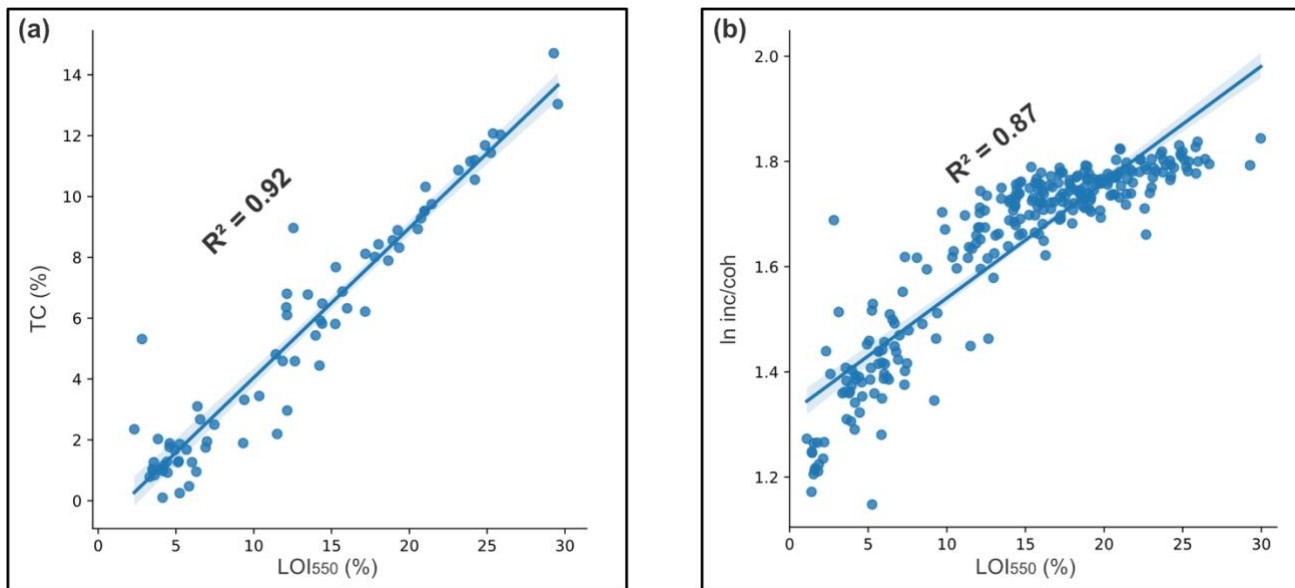

Figure 6 (a): Covariance plots with R$^2$ values for TC vs LOI 550º C and (b) for inc/coh vs LOI 550º C.

As expected, LOI$_{550}$ and BiSi reach their minimum in Units T5 and F, as well as with identified tephra layers. Furthermore, allochthonous proxies increase towards Unit B, while organic proxies display opposite trends (Fig. 3). LOI$_{550}$ shows a moderate decrease from the bottom of LME-CP to the base to Unit B (except for low values within the crystal-bearing Unit G). At Unit B they reach lowest values and abruptly increase again at the base of Unit A towards the surface. Maximum values are found in Unit H and in surface sediments (Unit A).

## 5 Interpretation

### 5.1 Proxy Analysis

The high correlation of Ca, Ti, K and Sr for LME-CP (Fig. 7a) are compatible with geochemical data of the Meseta Chile Chico basalts (De La Cruz and Suárez, 2008; Espinoza et al., 2005) pointing to these geological formations as the main source of clastic sediment components.

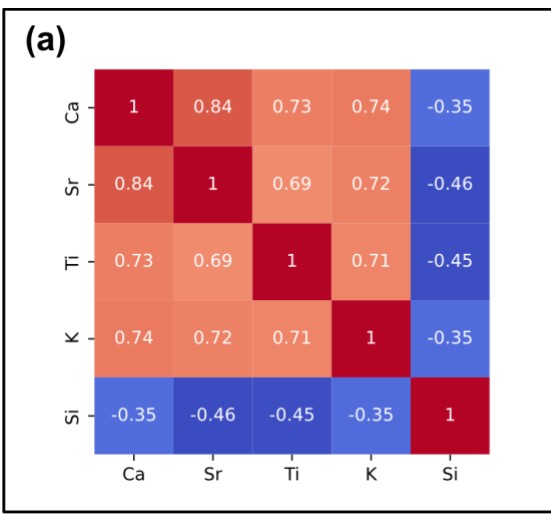
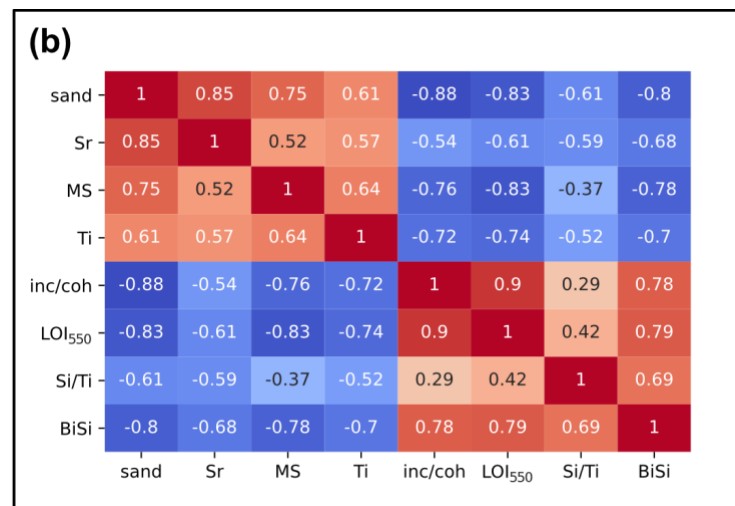

Figure 7 Correlation matrix for (a) elemental clr values (n=2,719) and (b) selected proxies (n=146).

Ti commonly used as proxy for catchment erosion (Davies et al., 2015) correlates well with MS (R=0.64, Fig. 7b). We associate this correlation to Ti-Fe oxides such as ilmenite and magnetite present in the groundmass of Meseta Chile Chico basalts (Espinoza et al., 2005) and consequently, we interpret Ti values as proxy for influx of (chemically) unweathered basalt clasts from the catchment.

Similarly, we found a strong match between Sr and sand percentages (R=0.85), indicating that the sand fraction is enriched in Sr. We attribute this to high concentrations of Ca-minerals (feldspars), since Ca and Sr (R = 0,84) are usually components of the same mineralogy with the exception that Ca is also related to most tephra layers (only T3 is depleted in Ca) and mixture of glass components with the sediments, whereas Sr is depleted in most volcanic ashes. For this reason, we choose Sr instead of Ca as a proxy for Ca-bearing silicates.

To assess the source variability of allochthonous sediments, we use the Ca/Ti ratio. We observe that this ratio remains mostly invariable between Units H and C (except for Unit T5). Thus, the source of siliciclastic material remained constant from the base of LME-CP upwards to 60 cm composite depth. Similar to the Sr values, high Ca/Ti ratios (Fig. 3) between 60 and 25 cm suggest that the source of Ca for this section is more related to Ca-silicates, most likely Ca-feldspars rather than to remobilization of fall-out deposits of T2 (enriched in Ca, according to our XRF data).

We also compare the proxies for minerogenic components (Ti, Sr, Ca/Ti) with the minerology of Unit B, where allochthonous parameters reach their maximum. The minerogenic composition of LME sediments, as observed for its sandy fraction, reflects high concentrations of sand-sized clastic material, such as quartz, feldspars and dark (mafic) volcanic lithics. This correlates well with the insights presented above and validates our chosen ratios for allochthonous proxies.

The organic proxies $LOI_{550}$, inc/coh and BiSi also show strong correlations amongst each other (R=0.78-0.9, Fig. 7b). Furthermore, to evaluate the overall Si content of the sediments, we tested Si/Ti as a proxy for BiSi percentages (Melles et al., 2012) and determined a good correlation between both parameters (R = 0.69). This documents that elemental Si variations with depth are mostly influenced by diatom accumulation, although siliciclastic detritus including pyroclastic fragments occur to a lesser degree.

## 5.2 Holocene paleoenvironmental evolution of Meseta Chile Chico

To evaluate the change of depositional conditions we plotted selected proxies vs time (Fig. 8). Ca/Ti, Ti and Sr, which represent detrital source variability, basaltic input, and sand-sized clastic supply, respectively, are considered as indicators for allochthonous input. Based on the significant correlation with BiSi as well as with $LOI_{550}$ (Figs. 3, 6b, 7b), we interpret the inc/coh ratio as an indicator for lacustrine productivity (Fig. 8). Variability of these proxies during the last 10,000 years suggests four major stages of lacustrine sedimentation.

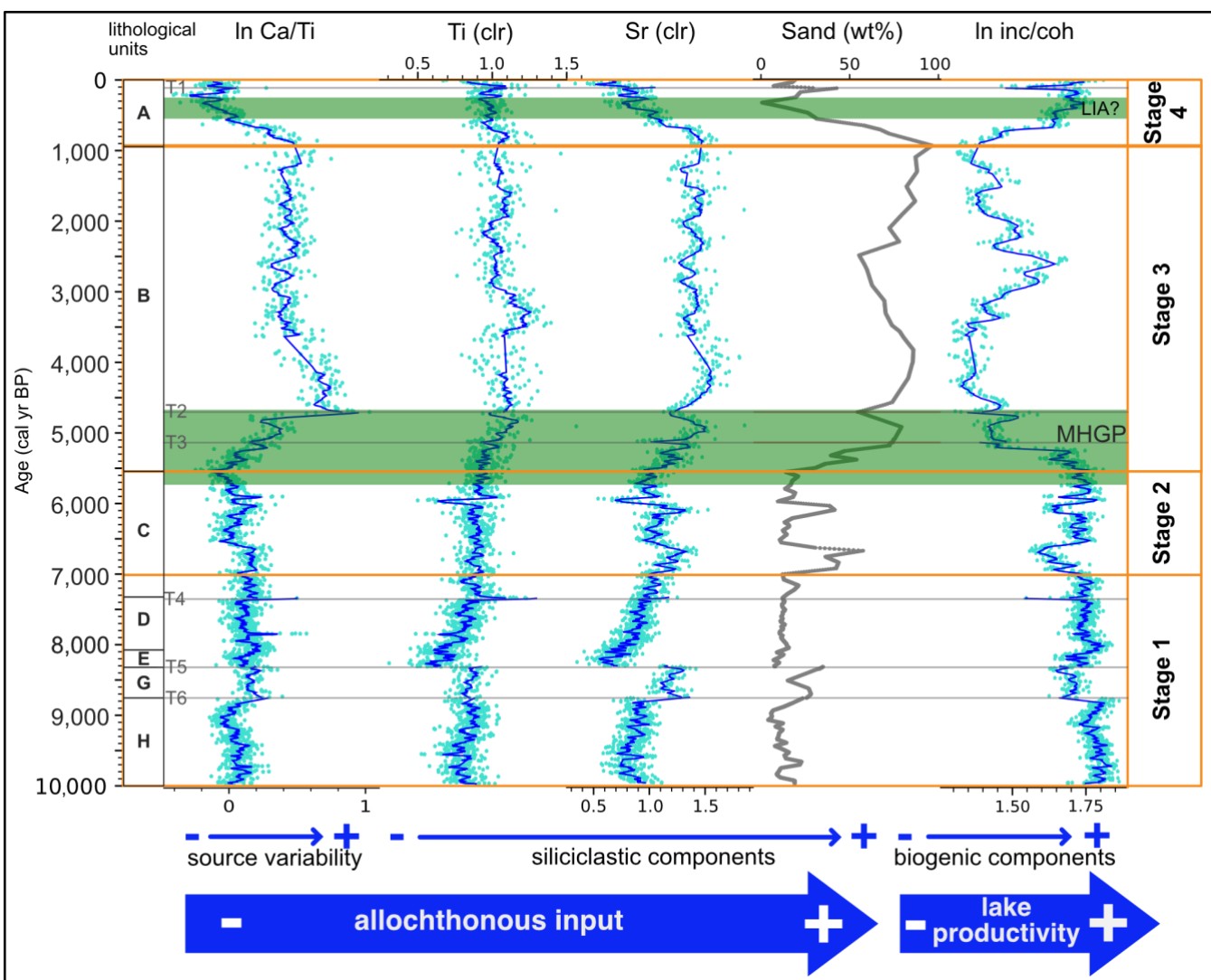

Figure 8 Lithology, running mean (dark blue line) and raw data (light blue dots) for selected element ratios, sand content and inc/coh ratio vs age and their interpretations for LME-CP. Grey lines labelled as T6 to T1 indicate instantaneous tephra deposition. The Mid-Holocene glacial period (MHGP) according to glacier chronologies within our study area (for references: Fig. 9) and the Little Ice Age (LIA) are marked.

The oldest stage recorded (Stage 1) started at least 10,000 years ago. It is characterized by high lacustrine productivity, which is decreasing with time. Consequently, allochthonous proxies have low values but increase slowly during Stage 1. This trend is interrupted around 7,000 cal yr BP by a sudden increase of allochthonous proxies between 7,000 and 6,500 cal yr BP, and between 6,200 and 5,900 cal yr BP. Rising values of Ti point to the Meseta Chile Chico basaltic units (Fig. 1c) as main detrital source of the sediments. Small variations in the Ca/Ti ratio are likely due to the presence of pyroclastic material (reworked from T4, T5 and T6, Fig. 3) rather than changes in detrital provenance. Accordingly, we interpret Stage 1 as a relatively stable period of high lacustrine productivity until 7,000 cal yr BP, when this development was interrupted by short periods of higher clastic supply from the catchment, marking the beginning of Stage 2 and the transition into completely different environmental conditions of Stage 3 (Fig. 8).

An abrupt increase in terrigenous components takes place at ~5,500 cal yr BP, setting the start of Stage 3. An enhanced deposition of terrigenous material dilutes biogenic production, causing organic components to drop as documented by a decreasing inc/coh ratio.

Besides an exceptional peak of organic components around 2,500 cal yr BP, Ti values indicate that detrital supply peaked at ~3,400 cal yr BP and remained relatively high with some oscillations throughout the entire Stage 3. Sand percentages and Ti

show an overall high correlation for the Early Holocene, but start to diverge after 4,700 cal yr BP. At 1,000 cal yr BP sand-sized components and Sr values reach their maximum. These differences together with the geological setting of the Meseta Chile Chico points to a concentration of Ca (Sr)-feldspar minerals within the sand fraction caused by stronger erosion of the Meseta Chile Chico basalts or by a change in the main detrital source from alkali-enriched Upper Basalts to the more mafic (non-alkaline) Lower Basalts of the Meseta Chile Chico. This suggest that the catchment of LME might have extended into the ridge that connects Meseta Chile Chico and Cerro Las Nieves (Fig. 1c) during this stage.

Around ~800 cal yr BP, terrigenous supply starts to decrease, and autochthonous lacustrine deposition (inc/coh ratio) predominates again, setting the start of Stage 4 and indicating a major change in sediment supply. Sand percentages decrease, while clay slightly increases (Fig. 3). At ca. 400 cal yr BP, biogenic components reach the maximum of Stage 4. Meanwhile, siliciclastic components reach their minimum around 400-300 cal yr BP and remain at high levels until present times. The proxy peaks at ~50 cal yr BP are related to remobilization of T1 and not to changes in paleoenvironmental conditions. This probably adds to a higher lacustrine productivity and points to a waning of runoff during the last millennium.

## 6 Discussion

### 6.1 Holocene paleoenvironmental evolution of the Meseta Chile Chico

Available chronologies of post LGM moraines (Fig. 9) in the vicinity of Meseta Chile Chico indicate that the Patagonian Ice Sheet must have retreated from surrounding valleys at 17.1-12.7 ka between latitudes 46-48º S  (Boex et al., 2013; Davies et al., 2018; Smedley et al., 2016; Turner et al., 2005). However, radiocarbon ages of deglaciation of regional lakes are younger than those indicated by their nearby moraine chronologies. Lake cores from the Río Cochrane valley indicate that sedimentation begun at 11,500 cal yr BP (Maldonado et al., 2022), while moraine studies in the nearby area indicate ages between 16.5-15.6 ka for ice retreat (Boex et al., 2013; Turner et al., 2005). We attribute this difference to a temporal gap between glacier recession and the subsequent beginning of organic sedimentation in lacustrine basins. Additionally, recent studies on glacial lake outburst floods date the opening of the valley of Río Baker, the event that separated the Parque Nacional Patagonia icefield from the Patagonian Ice Sheet, to around 12.8 ka (Benito and Thorndycraft, 2020; Thorndycraft et al., 2019). The only available environmental reconstructions for the Parque Nacional Patagonia icefield correspond to two moraine ridges near Fachinal (30 km NW of LME) with mean exposure ages of 8.2 and 6.2 and a related estimated paleo-equilibrium line altitude (paleo-ELA) of ~1,100 m a.s.l. (Douglass et al., 2005a). However, there is large scatter as ages from two neighbouring recessional moraines, i.e. 20.3-9.4 ka for the older moraine and 11-5.8 ka for the younger moraine, were treated individually rather than using a calculated mean (Davies et al., 2020). Our results show that at ca. 10,000 cal yr BP (minimum age for the formation of LME at 1,457 m a.s.l.) the paleo-ELA was already above 1,460 m a.s.l. and remained above this altitude throughout the Holocene. In consequence, we consider a maximum age of 20.3 ka for the ~1,100 m a.s.l. paleo-ELA as a maximum age for deglaciation of the Parque Nacional Patagonia Range and the Meseta Chile Chico topographic high. This indicates that Laguna Meseta was formed between 20 and 10 ka. The timing of Laguna Meseta, its altitude, and location with respect to the 10 ka paleomargins of the Parque Nacional Patagonia icefield (Davies et al., 2020), suggest that the formation of this lacustrine basin might be associated to either paraglacial or periglacial processes.

Reconstructions of deglaciation indicate a glacier retreat for the Early Holocene (11.5-8.0 ka), which triggered the drainage of several major regional lacustrine basins, such as Lago General Carrera, Lago Cochrane and the Tranquilo paleolake (Fig. 10a, Benito et al., 2012). Furthermore, the expansion of *Nothofagus* forests along the Chacabuco and Baker valleys and in the Lago Cochrane area (Villa-Martínez et al., 2012; Iglesias et al., 2018; Henríquez et al., 2017; Maldonado et al., 2022) for this period (Fig. 10d-e), points to more temperate or wetter environmental conditions. These correlate well with the maximum rates of lacustrine production documented by Stage 1 of LME. For the same time, moraine deposits are documented at ca. 9.5 ka for Laguna San Rafael (Harrison et al., 2012), at 9.8-6.7 ka for Cerro San Lorenzo (Sagredo et al., 2021) and at 11-5.8 ka for

Fachinal (Fig. 9, Davies et al., 2020; Douglass et al., 2005a). Nonetheless, these chronologies are either too scattered or they only represent local glacier advances to be regarded as regional glacial stages, explaining why they are not documented at Meseta Chile Chico.

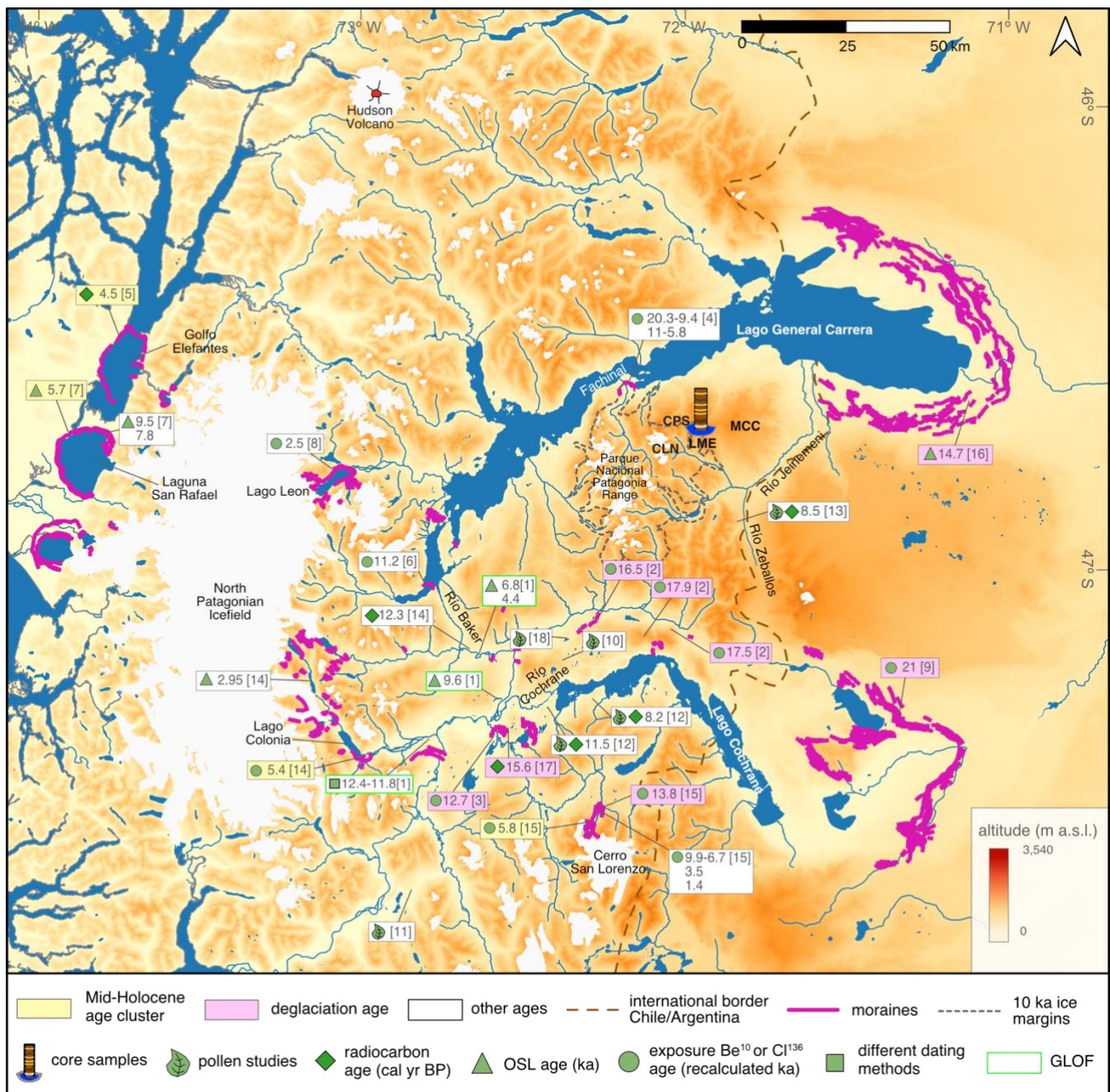

Figure 9 Ice retreat chronologies for the area east of the North Patagonian Icefield with the type of dating indicated. Lakes, ice cover, ice paleomargins, rivers and moraines are from Davies et al., 2020. GLOF: glacial lake outburst flood. Numbers in brackets refer to related publications: [1] Benito et al., 2021; [2] Boex et al., 2013; [3] Davies et al., 2018; [4] Douglass et al., 2005a; [5] Fernández et al., 2012; [6] Glasser et al., 2012; [7] Harrison et al., 2012; [8] Harrison et al., 2008; [9] Hein et al., 2010; [10] Henriquez et al., 2017 ; [11] Iglesias et al., 2018; [12] Maldonado et al., 2022; [13] McCulloch et al., 2017; [14] Nimick et al., 2016; [15] Sagredo et al., 2018, 2021; [16] Smedley et al., 2016; [17] Turner et al., 2005; [18] Villa-Martinez et al., 2012.

In contrast, the development of periglacial conditions above ca. 1,000 m a.s.l. documented by pollen studies from the Río Zeballos valley (Fig. 9) between 7,500-6,500 cal yr BP (McCulloch et al., 2017) is synchronous with the increase of sand at

7,000 cal yr BP that marks the beginning of Stage 2 at LME. This is pointing to wetter and/or colder climatic conditions. Records of paleovegetation at lower altitudes around the North Patagonian Icefield do not show periglacial evidence (Iglesias et al., 2018; Villa-Martínez et al., 2012). Moreover, Laguna Anónima documents an increase of *Nothofagus* at this time (Maldonado et al., 2022). In consequence, we associate the periods from 7,000-6,500 cal yr BP and from 6,200-5,900 cal yr BP with high sand/silt sedimentation to more humid conditions rather than to lower temperatures. These periods represent constrained events that are recorded only at higher and environmentally more sensitive areas.

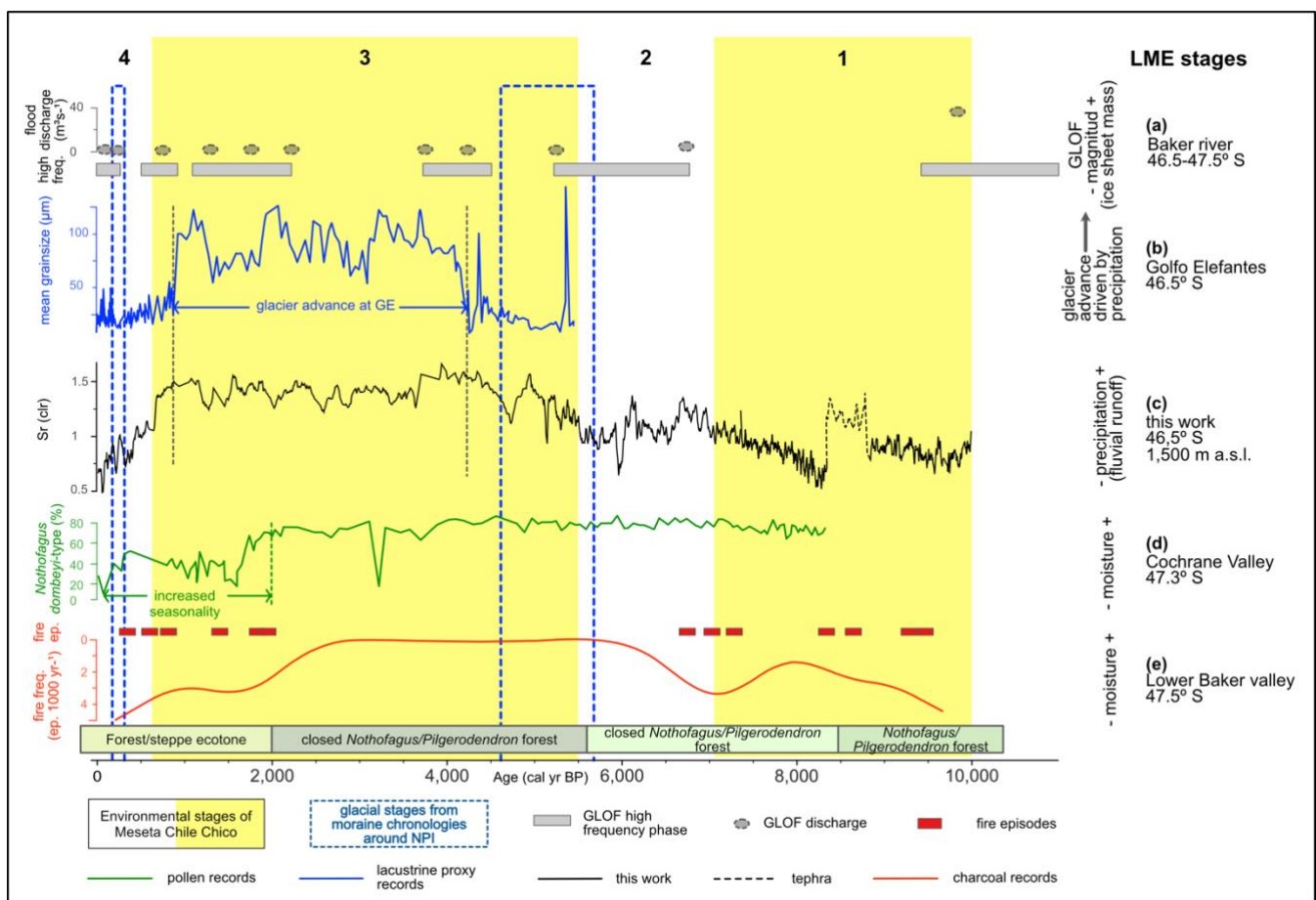

Figure 10 Selected proxies from LME in comparison to local records. (a) Magnitudes and frequencies of glacial lake outburst floods (GLOFs) (Benito et al., 2021). (b) Mean grainsize values at Golfo Elefantes (Bertrand et al., 2012) indicating an advancing Gualas glacier. (c) Sr (clr) values as proxy for fluvial runoff at Meseta Chile Chico. (d) *Nothofagus Dombeyi*-type tree pollen percentages from lacustrine records of the Lago Cochrane Valley as proxy for moisture/precipitation (Maldonado et al., 2020). (e) Pollen zones, fire frequency (number of fires per 1,000 years) and fire episodes at Lower Baker Valley (Iglesias et al., 2018). Stages from Fig. 8 (white yellow bands) and neoglaciations around the North Patagonian Icefield (squares with dashed blue lines) are also displayed. Ep.: Episodes; Freq.: frequency.

Contrarily, the younger Middle to Late Holocene (ca. 6-0.2 ka) moraine dates show active advances and retreats of the ice cover throughout Central and South Patagonia (Davies et al., 2020; Glasser et al., 2005; Reynhout et al., 2021). The transition from autochthonous to allochthonous deposition is observed for LME-CP between 5,500 and 4,600 cal yr BP (Fig. 8). This timing coincides with ages between 4.5 and 5.7 ka determined for moraines in the vicinity of LME (Figs. 9 and 10, Fernández et al., 2012; Harrison et al., 2012; Nimick et al., 2016; Sagredo et al., 2018).

Predominance of terrigenous sediment supply is recorded at LME until 900 cal yr BP (end of Stage 3). Thus, erosion and fluvial transport of clastics dominated on Meseta Chile Chico. Along the western slope of the North Patagonian Icefield, at Golfo Elefantes, sediment records document similar trends for grainsize variability between 4,800 and 850 cal yr BP (Fig.

10b-c), which are attributed to large precipitation-driven advances of the Gualas glacier (Bertrand et al., 2012). The correlation between both records suggests that more precipitation was responsible for higher influx of clastic and coarse (sandy) sediment to LME. We consider that the slight temporal offset between LME (5,500-900 cal yr BP) and Golfo Elefantes (4,800-850 cal yr BP) may be a consequence of asynchronous climate responses related to their relative positions with regard to the main Andean Range axis and the effect of calving processes that control west flowing glaciers at the North Patagonian Icefield (Harrison et al., 2012). However, such differences might as well be explained by dating uncertainties.

Humid conditions are documented by predominance of closed forests and low fire activity in the lower valley of Río Baker (Fig. 10e) and in Río Zeballos at ca. 5,800 and 5,300 cal yr, respectively (Iglesias et al., 2018; McCulloch et al., 2017). The high sensibility of these sites and of the Meseta Chile Chico may be attributed to their high-altitude positions, and thus their enhanced exposure to the wet and/or cold conditions that triggered the expansion of Middle Holocene glaciers. Meanwhile, pollen reconstructions from other sites around the North Patagonian Icefield do not indicate significant environmental changes around this time. In fact, they reveal discrepancies with a decline in the *Nothofagus* forest accompanied by a higher magnitude and frequency of fire events between 3,800 and 2,000 cal yr BP (Maldonado et al., 2022). Further north, a similar trend is documented in the Río Cisnes Valley (Fig. 1b), where the *Nothofagus* forest underwent a retraction around 4,200 cal yr BP (De Porras et al., 2012; De Porras et al., 2014). This regional environmental signal for the Late Holocene has been associated with marked seasonality, but it may also have been influenced by fluctuations in human occupation along these valleys (De Porras et al., 2014; Maldonado et al., 2022; Méndez et al., 2016). Nevertheless, whether strong climatic variability at an interannual-decadal scale played a major control in sedimentation on Meseta Chile Chico eludes the scope of our sediment records.

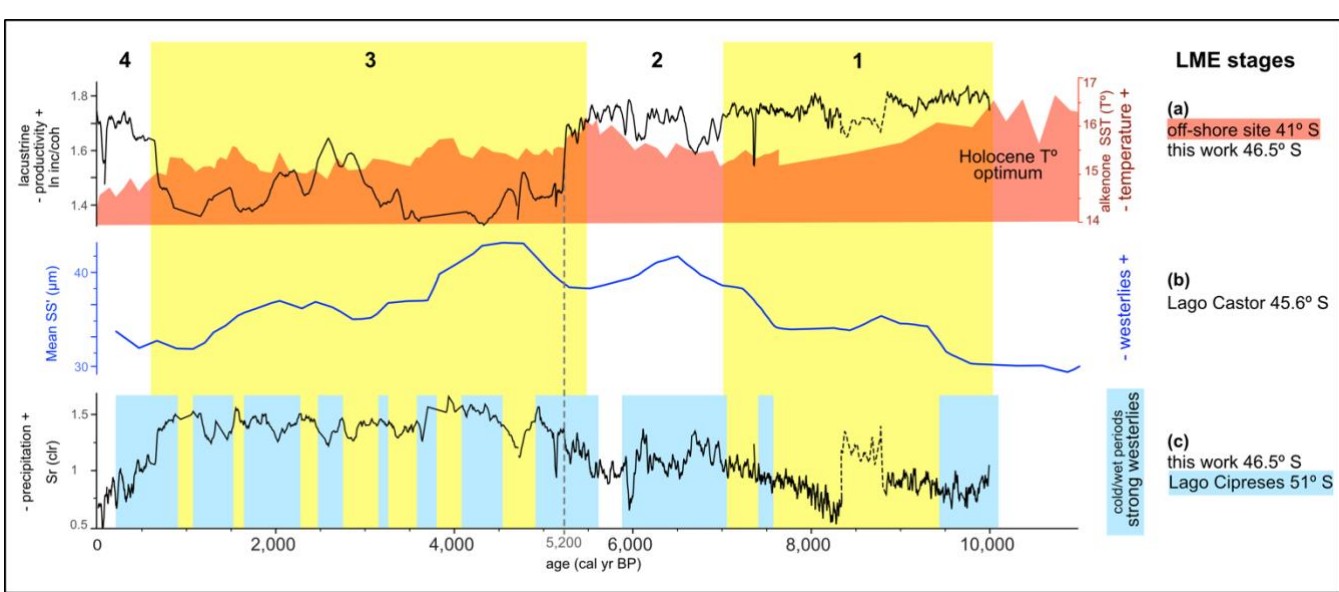

Figure 11: Proxies from LME in comparison to regional paleoclimate proxies. (a) Paleotemperatures according to an alkenone SST record (red) from marine sediments at 41° S (Lamy et al., 2002; Kilian and Lamy, 2012) and lacustrine productivity from LME (black). (b) Mean of modified sortable silt values (10-125 μm: SS') of lake sediments from Lago Castor as proxy for bottom current and westerly wind intensities (Van Daele et al., 2016). (c) Sr clr values (black) as proxy for fluvial runoff at Meseta Chile Chico and cold and wet periods (light blue bands) based on the non-arboreal pollen record from Lago Cipreses (Moreno et al., 2018). Symbology is displayed in Figure 10.

Local studies on climate conditions do not constrain Holocene temperature variability and therefore we consider temperature reconstructions based on marine sediments, which are the only data available for Patagonia. Marine records from 41° S (Kaiser and Lamy, 2010; Lamy et al., 2007) and from South Patagonia at 53° S (Caniupan et al., 2011) show little correlation with our

data (Fig. 11a). However, the change from warmer to colder thermal conditions recorded by sea surface temperature is synchronous to the strong decrease of lacustrine productivity at ca. 5,200 cal yr BP, suggesting that temperature had a relevant control on the transition from Stage 2 to Stage 3. Because ocean temperature reconstructions from the Pacific do not necessarily reflect thermal conditions of the eastern slope of the Andes, we consider that the influence of temperature at Meseta Chile Chico remains arguable. In contrast, our sedimentary records show a better correlation with different moisture proxies from our area throughout the Holocene (Fig. 10), pointing to precipitation as the main driver for this period rather than cooler temperatures.

To understand the relationship between our record and the SWW throughout the Holocene, we compare with proxies from nearby sites, including sites located east of the main Andean Range axis. Studies based on modern climate records indicate that leeward (east) of the Andes stronger westerlies induce lower moisture and higher evaporation and, therefore, are associated with dry periods (Garreaud et al., 2013) and lake level low stands (Kilian and Lamy, 2012). The opposite trend is documented west of the Andes with a positive correlation between wet conditions and westerly wind activities.

We observe that the correlation of SWW estimations between Lago Castor (Van Daele et al., 2016) and our record is not straight forward (Fig. 11b). Nonetheless, maximum wind intensities from Lago Castor and highest precipitation around the Middle Holocene at LME are synchronous. Meanwhile, SWW reconstructions from Lago Cipreses (Fig. 11c, Moreno et al., 2018) suggest a predominance of phases of strong westerlies with intensified precipitation between 7,000 and 5,900 (Stage 2), and between 5,500 and 900 cal yr BP (Stage 3) in the Meseta Chile Chico.

In consequence, periods of high precipitation at LME seem to have been mainly driven by high SWW activity associated to their northward expansion (Lamy et al., 2010) bringing cold/wet phases into the middle latitudes, which started around 7,000-8,500 cal yr BP but became predominant around 5,000 cal yr BP. The frequency of these intervals was mainly controlled by changes on a millennial to centennial scale of the Southern Annular Mode (SAM, Moreno et al., 2018). Therefore, our findings imply that the SAM-driven climatic conditions that triggered neoglaciations in Antarctica and Southern Patagonia (Kaplan et al., 2020) extended as well into Central West Patagonia.

Additionally, the straightforward relation between precipitation at Meseta Chile Chico and the SWW indicate that the eastern slope of the Andean range was a transition zone between the humid Andes and the dry steppe region, over which this wind belt still retained moisture from the Pacific Ocean during the Early-Middle Holocene.

After 1,000 cal yr BP, LME sediments document a shift back to higher accumulation of biogenic components with less clastic input. This tendency continues until 300 cal yr BP. Predominance of silt and clay for our record (Unit A) refers either to a moderate erosion intensity uncapable of transporting sand but enough to cause watershed erosion, and/or to enhanced influx of clay-silt sized minerogenic clasts, such as micaceous minerals, caused by higher chemical weathering of rocks that occupy the catchment of LME.

Between 300 and 200 cal yr BP, inc/coh ratio reaches high values and, in consequence, a strong drop is observed in elemental data (Ca/Ti, Ti and Sr) as well as in sand percentages (Fig. 8). The age of this signal overlaps with the Late Holocene glacial period known as the "Little Ice Age" (Davies et al., 2020) and widely documented within our study area (Davies and Glasser, 2012; Garibotti and Villalba, 2017; Nimick et al., 2016; Sagredo et al., 2021). However, a high lacustrine productivity signal suggests that favourable and more temperate rather than cold/wet conditions controlled the environment at Meseta Chile Chico. This points to a climate signal being overprinted by local sedimentation dynamics at Meseta Chile Chico, and that low allochthonous input was mainly responsible for high lake production at LME at this time. However, the signal of our record could also be the result of the resolution of our age-depth model for the Late Holocene (2000-0 cal yr BP). We expect to address these paleoenvironmental issues with more detail for this period with investigation of another lake record from Meseta Chile Chico.

Although human presence east of the North Patagonian Icefield became regular around 3,000 cal yr BP, dated archaeological sites along the SE border of Meseta Chile Chico point to human occupation between 500 and 1,560 cal yr BP (Nuevo-Delaunay

et al., 2022). However, preliminary archaeological surveys across the plateau revealed small surface concentrations of lithic material indicating the procurement of toolstones, signalling minor evidence of anthropogenic influence on Meseta Chile Chico during the Holocene (Méndez et al. 2023). Therefore, we assume that also the younger section of LME-CP (Stage 4) reflects mainly natural Holocene environmental variability and not human impact.

## 6.2 Sedimentary environments at Meseta Chile Chico throughout the Holocene

According to glacial reconstructions within our study area, the glaciers of the Parque Nacional Patagonia icefield that covered the Meseta Chile Chico during the LGM had already begun their retreat before 10 ka (subchapter 6.1). They also indicate that glaciers at other sites from West Patagonia mostly continued retreating until ca. 6 ka and that they presented several advance/retreat cycles between 6 and 0.2 ka (Davies et al., 2020; Benito et al., 2021; Glasser et al., 2005). Therefore, we consider the possibility that the easternmost glaciers of the Parque Nacional Patagonia icefield advanced and retreated several times from the Meseta Chile Chico before permanently abandoning this catchment system in the Holocene. Moreover, the extension of the basin and catchment of LME and the Chile Chico Meseta drainage system must have been largely affected by geomorphological responses to glacial dynamics during this time.

Maximum lake productivity (inc/coh ratio) recorded at the lowermost layers of LME sediments (Stage 1), suggest that its oldest sediments were accumulated when LME was already a distal lake basin with respect to the glaciers of the Parque Nacional Patagonia icefield. Furthermore, the fine grainsize of the minerogenic components (high silt/sand ratios) and the lamination (Units C, D, E, G and H, Fig. 3) of Stages 1 and 2 are characteristic of fluvioglacial settings with silt-sized sediment being transported by downwashing and reworking of freshly exposed glacial deposits and accumulated downslope by particle settling in lake basins (Ballantyne, 2003).

Glacial conditions on Meseta Chile Chico that might have persisted until the Middle Holocene agree with the abrupt switch from autochthonous to allochthonous sedimentation recorded at the beginning of Stage 3. If we assume a distal glacial source for Stages 1 and 2, then it is also possible that the Middle Holocene glacier advance recorded by moraines around LME might have triggered an eastward advance of the outlet glaciers of the Parque Nacional Patagonia icefield into Meseta Chile Chico and, therefore, enlarging its catchment area. Former studies on glaciolacustrine sediments associate the occurrence of laminated sandy layers with historically recorded glacial stages (Leemann and Niessen, 1994), indicating that such an advance could be the source of sand clasts and additional detrital signals in the sediments of Stage 3. The occurrence of a Middle Holocene glacial advance in our studied plateau is also supported by regional proxies, which indicate lower temperatures and a higher precipitation regime for Central Patagonia during this period (Figs. 10 and 11, Bertrand et al., 2012; de Porras et al., 2012; Iglesias et al., 2018; Kilian and Lamy, 2012).

As geomorphological mapping shows (Davies et al., 2020; this work), even after the Parque Nacional Patagonia icefield had retreated from Meseta Chile Chico, smaller glaciers remained around the summit of Cerro Pico Sur for a longer period (Fig. 1c). These glaciers might have been responsible for second order oscillations within our data during Stage 3, since restricted glacial advances are documented around the North Patagonian Icefield between ca. 4.3-1.2 ka at Lago Colonia, Lago León, Cerro San Lorenzo and Golfo Elefantes (Bertrand et al., 2012; Harrison et al., 2008; Nimick et al., 2016; Sagredo et al., 2021) as well as at other sites from South Patagonia (Hall et al., 2019; Reynhout et al., 2019; Strelin et al., 2014).

It is impossible to estimate when glacial influence ceased on Meseta Chile Chico during the Holocene. This event would have triggered the transition of LME from a fluvioglacial into a fluvial basin. Nonetheless, the steady increase in sand/silt ratios between ca 2,500 and 900 cal yr BP might be indicative of the settling of a mature fluvial system after paraglacial reworking ended on the plateau.

It is likely that also aeolian transport variably contributed to sediment accumulation at LME. Sedimentation of fine sediment originating from deflation of recently exposed glacial deposits after ice recession is a characteristic process within paraglacial

environments (Ballantyne, 2003). Accordingly, high silt/sand ratios of Stages 1 and 2 could have a secondary eolian component. In contrast, sediment layers for Stage 3 are sand dominated. Because sand can only be wind-transported by saltation, wind driven accumulation of sediments in Stage 3 would have required a proximal constant source of sand sized detrital material. We consider it as unlikely that such a source could have developed in an isolated high-altitude area, dominated by volcanic rocks such the Meseta Chile Chico. Moreover, small fluctuations of sand percentages in LME within Stage 3 do not correlate with SWW intensity records from Patagonia. Therefore, within our studied area, wind sedimentation was probably overprinted by glaciofluvial and/or fluvial processes strongly linked to the rapidly changing wet/dry conditions that controlled landscape evolution in Patagonia after deglaciation.

At the end of Stage 3 (900 cal yr BP), LME changed back to high lake productivity and high silt/sand ratio conditions, which remained relatively stable until modern times (Stage 4). This suggests that around this period the sedimentary environment of Meseta Chile Chico evolved into the low gradient and ephemeral streams that currently control sediment supply at LME and nearby lakes. Since regional proxies and the results from this study are non-conclusive for precipitation/temperature variability around this time (Fig. 10), we attribute these changes to catchment morphology, weathering and slope gradients.

Sediment changes within our records are consistent with two scenarios: i) a fluvial basin mainly controlled by precipitation-driven changes with superficial runoff; ii) a glacial or periglacial setting, controlled by episodes of glacier advances and retreats that were induced by both thermal (at least until the Middle Holocene) and precipitation oscillations, until a fluvial system was established sometime after 5,500 cal yr BP. Nonetheless, it is most likely that these environments coexisted and also alternated throughout the Holocene on Meseta Chile Chico. Additionally, ice reconstructions for plateau icefields are challenging since evidence of glaciation is commonly scarce. In these settings, glacier dynamics are directly dependent on plateau area, main wind direction and altitude (Rea et al., 1998; Rea and Evans, 2003). Thus, they can be asynchronous compared to respective valley glaciers. Therefore, due to the lack of moraine chronologies for Meseta Chile Chico, the existence of glacial advances after the LGM remains speculative.

## 7 Conclusions

The sediment record of LME started before ~10,000 cal yr BP, when ice of the LGM retreated from Meseta Chile Chico. With the exception of two isolated wetter periods (7,000-6,500 and 6,200-5,900 cal yr BP), glacier recession caused by temperate and drier conditions continued between 10,000 and 6,000 cal yr BP, delivering low amounts of minerogenic components into the lake. Environmental conditions for accumulation of biogenic lacustrine sediments were optimal during this period but steadily decrease. Around 8,300 cal yr BP, fall-out deposits originating from the H1 eruption of Hudson volcano reached the lake with deposition of primary and remobilized tephra.

At 5,500 cal yr BP, there was a notable shift in sedimentation in LME from organic to allochthonous minerogenic material. This shift was primarily driven by increased fluvial erosion of basaltic rocks, which, in turn, were influenced by higher precipitation associated with stronger SWW activity and glacial advances in Patagonia.

Between 900 and 300 cal yr BP terrigenous input decreased again. At 300 cal yr BP sediment accumulation stabilized and was mainly controlled by lacustrine productivity without major changes until modern times. This indicates that the sedimentation of present day Meseta Chile Chico reflects similar environmental conditions like during the past 300 years.

Changes in granulometry and provenance of the sediments of LME are consistent with glacial advances and retreats in the catchment until the Late Holocene. However, since there are no moraine dates available for Meseta Chile Chico, it is not clear when glaciers finally retreated.

The lithological variability recorded by the lacustrine sediment record from Meseta Chile Chico, its altitude and proximity to the North Patagonian Icefield as well as the low degree of human influences provides a privileged setting for a detailed study of Holocene environmental conditions.

**The Supplement related to this article is available online at doi:xxxxxxxxx/cp-xxx-supplement**

685 **Data availability**

All analytical data of this study is accessible for downloading at https://doi.org/10.1594/PANGAEA.961940

**Author contribution**

The objectives of this study as well as the planning for sampling campaigns was carried out by C. Franco, A. Maldonado, M.E. de Porras, A. Nuevo-Delaunay, and C. Méndez. Analytical lab work was carried out by C. Franco, A Maldonado, C. Ohlendorf, 690 and A.C. Gebhardt. Lithological descriptions, age-depth model, and tephrochronology, as well as manuscript writing was performed by C. Franco. B. Zolitschka and A. Maldonado conceptualized this research. All authors contributed to writing and revising this manuscript.

**Competing interest**

The authors declare that they have no conflict of interest.

695 **Acknowledgements**

Thanks go to Rafael Stiens (GEOPOLAR, University of Bremen), Pascal Daub (AWI), Claudia Hernandez (CEAZA), and Dr. Philipp Böning (University of Oldenburg) for analytical support, to Stella Birlo (GEOPOLAR, University of Bremen) for assistance with python coding and to the Corporación Nacional Forestal (CONAF, Chile) for authorising our studies in protected areas of the Meseta Chile Chico.

700 **Funding**

This study is supported by the German Academic Exchange Service (DAAD, Research Grants – Doctoral Programmes in Germany), Agencia Nacional de Investigación y Desarrollo (Ministry of Science, Chile): ANID FONDECYT 1210042 and ANID Regional R20F0002 grants (Ministry of Science, Chile).

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
