# Peer review of "Holocene environmental and climate evolution of Central West Patagonia as reconstructed from lacustrine sediments of Meseta Chile Chico (46.5° S, Chile)"

_EGUsphere, 2023_

## Author Response (AR1)

**Authors reply to Anonymous Referee #1 (RC1)**
Text in blue corresponds to authors responses.
Text in red corresponds to changes made in MS

**Holocene environmental and climate evolution of Central West Patagonia as reconstructed from lacustrine sediments from Meseta Chile Chico (46.5º S, Chile)**
Franco et al., 2023

**1. General comments**

- A 3 m-long composite lake sediment record (LME-CP) was analysed from Laguna Meseta (LME-CP) in Central Western Patagonia at the eastern margin of the North Patagonian Icefield (NPI). Based on a multiproxy lake sediment analysis over the past ~10 ka, the authors discuss the sedimentation dynamic to reconstruct the glacial and environmental history of the area. The sedimentation dynamic was then correlated to environmental changes around the study area, specifically, regional glacial oscillation and paleoclimate proxies, with the objective of deriving insights into the Holocene climate variability in Central Western Patagonia. The authors discuss, infer, and conclude that the major environmental changes during the middle Holocene are mainly controlled by precipitation variability linked with the evolution of Southern Hemisphere Westerly Winds (SHWW).
The manuscript is well presented and structured, with fluent and precise language. Additionally, the problem, statement, and objectives are clearly explained. The manuscript provides results from a new location in the east of the NPI that contributes to improving the knowledge base concerning the evolution of the SHWW and adds relevant information to the understanding of past climate in the region. The evolution of the SHWW is a key component of the South American climatic systems. This manuscript contributes results that support a better understanding of the behaviour of the evolution of the SHWW through the Holocene and therefore, is within the scope of this journal. The title's manuscript clearly reflects the scope of the research, focusing on where the study was performed, the type of records found, and the period when is the environmental reconstructions were made. The abstract provides precise and complete information about the content of the manuscript, however, one suggestion for modification is, "3 m-long continuous sediment record" should be changed to "3 m-long composite sediment record" or just "3 m-long sediment record".

The methods area is clearly outlined and organised, and the description of the experiments is sufficiently complete and precise to allow for their reproduction. However, the construction of the "composite sediment records" is not totally explained in the methods sections, but this issue can be resolved with a brief explanation.

The manuscript's figures accompany the text well, and these have detailed descriptions, which allow for a better understanding of the problems in the study area context. The manuscript is well presented, and the article is very well structured. The problem statement is clearly explained, as are the objectives, area of study, and the method, the latter, permits the reproduction of the results. The manuscript presents a well pool of multiproxy results, which support the interpretation, and reaches adequate conclusions.

The interpretation of the geochemical data is clear and thoroughly discussed, and this allows the reader to appreciate how the authors constructed a discussion and reached the conclusion. The number of references is adequate; however, it is recommended that the authors check the format of the references. There is a minor change to make, for instance, the "Https://" is sometimes presented before the paper's doi and other cases not."

We appreciate the general positive review. The suggestion made by RC1 regarding the construction of the composite sediment record is addressed in the specific comments section.

**2. Specific comments**

- "Line 132 mentions that three sediment cores were collected, and one of them will be used in the future for pollen analysis. Thus, the lector assume that two cores were utilised to construct the "3 m-long composite sediment record" (Laguna Meseta, LME-CP). Then, it is mentioned that the composite sediment records were constructed with additional help of six visible tephra as marker horizons for correlations between core sections. I understood (or I assumed) how the LME-CP was constructed after reading the results (Fig. 3) and Table 2 (AMS radiocarbon dates). I suggest adding a brief explanation in section 3.1 to clarify how the LME-CP was constructed, for example, mentioning how many overlapping drives of sediment (how long) were cored with a xxx corer from a depth of c xxx producing a composite lake sediment of 300 cm depth (Fig. 3), and that correlations of the overlapping core sections were based on (lithological changes?), and the correlation among sediments cores were made by…etc.

1.1 We will add supplementary material illustrating how we established the final composite sediment record.

Detailed composite profile is provided in the Supplement Fig. S01.

1.2 We agree, unquantified XRF geochemistry is not sufficient to address volcanic sources. We will rephrase this.

To avoid confusion, sentence was deleted

- Only one tephra layer was geochemistry identified and compared with previous data from H1 (Hudson 1 eruption) and M1 (Mentolat 1). The results from this study show similar geochemical characteristics as T5 (H1, Hudson 1) (Table 1: Fig. 4). Then, the authors mention a mean age of 8,415 cal yr BP (Stern et al., 2016), which was drawn in figure 6 but was not included in the age-depth model display. My point here is to ask why the authors do not include H1 to display the age-depth model and completely apply the tephrochronology concept? I suggest writing a brief phrase explaining why H1 age was not included in the age-depth model. For example, an answer could be associated with any of the following: the limits of the tephra layer are not clear, the previous H1 age is too old/young compared with this study, or another motive. This explanation is very useful because tephrochronology is frequently not used in new data and/or in the other data we want to compare, and in my opinion, tephrochronology is underused."

1.3 The age of the Hudson 1 eruption is not included in the age-depth model because it has actually been dated for several locations, in both surface deposits and sediment-core samples from which different ages (within a reasonable range) have been obtained. For this reason, although we cite the age from Stern et al. 2016 (being the latest published chronology for this tephra), we do not include it in our age-depth model. Moreover, as RC1 states, the exact depth of the top boundary of this tephra within our records is not precise. Instead we compare the range of all available ages from other sediment cores with our age-depth model to highlight that published dates for H1 and our chronology are consistent with each other. A more detailed explanation of this matter will be included in the revised manuscript.

Explanation regarding Hudson 1 and its considerations for the age-depth model was enlarged

**3. Format comments**

Line 65, should say (Fig. 2a)
Line 66, should say (Fig. 1c)
Line 66, should say (….: De Cruz and Suárez, 2008)
Line 91, at the beginning of the manuscript the references are organised by alphabetical order, then by year? should be consistent throughout the ms.
Line 132, include the point after (Fig. 2a)
Line 133, should say (Fig 2b)
Line 263, should say (Fig. 1b)

Line 277, should say (Fig. 4a)
Line 284, should say (Fig. 4b)
Line 308, should say (Fig. 5a)
Lines 310, should say (Fig. 5b)
Line 332, should say (Fig. 3)
Line 352, should say (Fig.7a)
Line 358 and 375, should say (Fig. 7b), should be consistent throughout the ms. Please check.
Line 405, should say 2,500 cal yr BP.
Line 448, Nothofagus, italic
Line 500, Nothofagus, italic
Line 663, in some references "the https://" missing, should be consistent throughout the ms.

Changes were made, References were corrected

Numbers in black correspond to paragraphs from the review of RC2, text in blue corresponds to authors responses.

**Holocene environmental and climate evolution of Central West Patagonia as reconstructed from lacustrine sediments from Meseta Chile Chico (46.5° S, Chile)**
Franco et al., 2023

- "The manuscript "Holocene environmental and climate evolution of Central West Patagonia as reconstructed from lacustrine sediments of Meseta Chile Chico (46.5°S, Chile)" by C. Franco et al. focuses on the reconstruction of Holocene environmental changes in West Patagonia using a 3-m-long lacustrine sediment succession. Overall, although the manuscript is well written, I ask for major revisions to better constrain the aim of this study, to improve the structure of the manuscript, and to make data and results reproducible. Particularly the (tephro-)chronology needs a thorough revision and clearer discrimination of raw data and modelled data. The tephrostratigraphy is weak, but an improvement might be beyond the scope of this manuscript and is time consuming. However, based on the weak tephrostratigraphy a correlation of tephra layers in the lacustrine sediments with those known from other records should be more carefully addressed and discussed."

2.1 We value the overall positive review of the manuscript.

In relation to our tephrochronological insights, the analysis conducted on the thickest tephra layer within our records serves the specific purpose of supporting the radiocarbon-based chronology of our sediments. We agree with RC2 that a more in-depth analysis of volcanic deposits is beyond the intended scope of this manuscript, since our research specifically aims to draw inferences on Holocene paleoclimate.

However, we appreciate the suggestion to provide a more detailed explanation of our chronological insights for T5 and enhance the clarity of this aspect (details provided below).

Explanation of our assignation criteria of T5 to Hudson 1 event was enhanced

- "The abstract provides a short introduction and summarizes the main findings of the study, i.e. the environmental changes over the last c. 10,000 years. Although most of the abstract reads well and is coherent, it is not clear to me how the authors come to the conclusion in the last sentence that environmental conditions during the Holocene were mainly controlled by precipitation variability and oscillations of the Southern Hemisphere Westerly Winds. While this might be correct, in the text previously, changes in sedimentation characteristics were explained with a broader variety of environmental conditions, including shifts in lacustrine productivity and variations in allochthonous sediment input, likely caused by colder and/or wetter conditions. The reduction to one of these factors in the last sentence is surprising and difficult to follow. I suggest some minor rewording here and there to emphasize the role of precipitation changes and the Westerly Winds."

2.2 We will enhance the discussion of SHWW in the abstract for the upcoming revisions.

Insights of the SHWW fluctuations during the Holocene were enhanced

- "The introduction needs major revision. The first of a total of three paragraphs deals with a period that is not covered by the sediment succession from Meseta Chile Chico. The two sentences of the second section describe the state of research on Holocene climate variability and on glacial advance and retreat in the region. The third paragraph defines the aims of the study.

2.3 The initial paragraphs of the introduction are dedicated to point out that the Quaternary history of Patagonia has predominantly been elucidated through investigations of glacier chronologies. However, these chronologies entail limitations, particularly concerning their discontinuous time ranges. This information serves as a foundation to explain that, in contrast, studies focusing on lake sediments (e.g. our study), offer valuable insights as continuous records of paleoenvironmental variability. This is the main objective of our study, which we intend to clarify in the introduction.

- This introduction is completely ignoring the existing knowledge on Holocene environmental changes, as it is derived from numerous marine and terrestrial records in the region and its realm (including e.g. Zolitschka et al. 2013, and references therein). Not a single note is found to precipitation changes and oscillations of the Westerly Winds, which according to the abstract are the factors controlling environmental, climate (and glacier?) changes in the region. There is a distinct need to span here the broader frame for the purpose of the new study, the new data and their interpretation. Some of the missing information is provided later, in chapter 2.2 ("Previous paleoenvironmental reconstructions"). However, also this chapter contains a lot of information from the pre-Holocene, a period, which is not covered with the sediment succession from the Laguna Meseta. I suggest to streamline the content a bit (also with respect to the knowledge on oscillations of the wind systems) and move large parts into the introduction."

2.4 The paleoenvironmental data specific to our studied area is presented in the regional setting chapter, not as a primary focus of the study objectives, but rather to establish the foundation for our interpretations. The literature cited, particularly addressing deglaciation processes preceding the Holocene at our specific location, is crucial as it refines the timing of the Meseta Chile Chico deglaciation, holding implications for the onset of sedimentation in our studied lake.

The paleoclimate findings derived within the manuscript are designed to provide insights for climate variability in Central West Patagonia specifically (~42º - 48º S, ~71.5º - 75º W). The selection of comparative records considers:
1) Their proximity relative to our study site, to mitigate potential latitudinal climatic gradients
2) Their similarity to the local setting of our lake in terms of altitude, position relative to the Andean axis, type of proxies used, catchment properties, etc.
3) Exceptions were made for studies addressing regional events well-documented throughout a larger section of Patagonia. In these cases, records are mentioned as general references.

As an example, the record suggested by RC2 corresponds to Laguna  Potrok Aike, Argentina. This lake is located at 52ºS (70ºW, 116 m a.s.l.), Southern Patagonia. It "represents one of the few non-glacial and extra-Andean lake archives…" and exhibits dynamics primarily influenced by wind intensities (Zolitschka et al., 2013). Therefore, the context of this lake markedly differs to our studied lake, which corresponds to an Early Holocene basin located at ~1,500 m a.s.l. on the eastern margin of the Andean range. It was formed by processes related to glacial retreat during the las glaciation and it is currently fed by mountain streams (this work).
Additionally, climate oscillations in Patagonia are still subject of large debate as their timing and latitudinal synchronicity remain largely unclear (Kilian and Lamy, 2012; Moreno et al., 2018). Hence, to establish a direct correlation between environmental records from Central (this research) and South Patagonia (Potrok Aike) would require a much broader review, extending beyond the scope of our research.
Consequently, records from Zolitschka et al. (2013) as well as from other locations in Patagonia are not presented for direct comparison. Nevertheless, we will reevaluate available records for Central West Patagonia in order to complement the introduction chapter

Introduction chapter was complemented (see response 2.2)

- "In the chapter on the regional settings, I miss important information about the lake itself, e.g. lake size, what is the bathymetry based on (Fig. 2), what is the hydrology of the lake with respect to temperature, mixis, etc.? What is the trophic state, what is the dimension of the catchment area? What is the predominant vegetation in the catchment area and how dense is the vegetation cover?"

2.5 The sampling campaign took place in a remote area of the Andean Range (1,500 m a.s.l.). Therefore, limnological data such as water temperature profiles, trophic states and mixing regimes are not available. Data regarding catchment area, lake size and bathymetry will be included in the revised manuscript.

Missing information was included in MS. See "2.1 Geomorphology and modern setting" and "3.1 Bathymetry, sediment coring and subsampling"

• In the Methods and Materials chapter, please add some more information to the length of the individual core sections, and how many sections have been recovered at the individual sites. This is important to follow the correlation of individual core sections to a composite record. Adding a table with individual core lengths, field depths, sites and/or tools used would help to better understand the coring procedure and splicing of core sections to the composite record.

2.6 Regarding the composite profile, we consider adding supplementary material to illustrate the composite sediment record.

A supplement with composite diagram figure was added

• I also suggest to restructure this chapter, starting in line 138 with a subchapter on scanning and logging (XRF and mag sus), then geochemical analyses on tephra, followed by analyses on discrete samples (grain size and bulk geochemistry), before dating, age-depth modelling and handling with published data is described.

2.7 We agree on placing the subchapters about scanning ahead of all discrete sampling.

Order of subchapters was change in MS

• With respect to tephra layers, why was only one tephra selected for major and trace element analyses?

2.8 The T5 tephra layer was analysed because of its exceptional thickness (14.5 cm), very well defined lower boundary and the fact that it is highly enriched with pumice fragments. These characteristics provide a unique opportunity for tephra geochemistry without further treatment. The remaining five tephra layers have thicknesses that range between 8 and 17 mm and do not present high concentrations of large pumice fragments, they are mostly enriched in > 1 mm glass fragments. Their analysis would have required lab techniques to separate glass from sediment, which were not available for this study.

Moreover, descriptions and age range for this tephra, which are very well documented in several other studies match our samples and age-depth modelling. Tephrochronological records for later tephra deposition throughout the Holocene are less clear in the literature and would require extensive analytical work for determination.

A larger explanation on our tephra records and their analysis was provided in MS

• Can you provide some information how accurate WD XRF is for trace element analyses in lake sediments (particularly with respect to Sr, I assume that calcite precipitation in the lake is limited, but it would be good to discuss this a bit more, probably later in the text)?

2.9 XRF values for Sr are within 3.805-1130 counts/sec, similar to the values obtained for some major elements such as Ti (3.560-1.090 counts/sec). This is related to the detection capacity of the XRF method itself, which varies among elements depending on their atomic structure rather than on their concentrations (hence, the nonquantitative nature of the method). We agree to provide a more detailed description of the readability of the considered elements in the supplementary materials along with the description of the composite record.

As stated in line 305-306, total inorganic carbon (TIC) was measured and values below 1% were obtained for all tested samples. No calcite crystals or any other carbonate minerals were determined by microscopic observation. This data lead us to assume that carbonate precipitation in the lake is very low or even absent.

Raw data on element counts as obtained from the XRF method is provided to account their variability and validity

- With respect to chronology and radiocarbon dating, why was bulk sediment used for dating, if plant remains have been detected (chapter 4.1; Fig. 3)?

2.10 Although plant remains are present in the core samples it was not possible to assign them to a certain species. Hence, bulk sediment samples were used for dating, especially as carbonates (reservoir effects) are lacking.

In the MS It was specified that the observed plants were not possible to identify

- The subchapter "Tephra records and tephrochronology" (pls reword; see below) needs further elaboration. Tephrochronology is not the right wording here, as tephra was not used to constrain the age model (at least from what I understood when reading the text, however, see also comments below). In contrast, the age model was used to assign one of the tephra layers to a known eruption. Overall, I miss further information on (1) why are Cay and Maca volcanoes not included in the discussion?

2.11 As stated in lines 261-264, "This age suggests a likely correlation with tephra records from Mentolat (Men1) and Hudson (H1) volcanoes…", the discussion is built upon previously published tephra records with ages and distributions potentially correlating with the T5 tephra. Macá and Cay volcanoes are excluded because their deposits have not been recognized south of Lago General Carrera (southernmost deposits have been described at ~45.5º S, Weller et al., 2017). Moreover, the only large eruptive event recognized for Macá volcano corresponds to a Late Holocene event (Weller et al., 2017; Naranjo and Stern, 2004). This event is too young to be correlated with T5. For Cay volcano no large regional events are documented. In fact, previous publications suggest that it did not have important activity during the Holocene (Weller et al., 2015; 2019).

We will include this brief insight about Macá and Cay volcanoes in the discussion of tephra deposits for the revised manuscript.

Macá and Cay volcanoes were included in the discussion. The whole subchapter was improved with details.

- ….What is H0-H3 shown in Fig. 4? Do all dots in Fig 4b represent bulk samples? Why are there different markers (dots, diamonds, triangles) used in Fig 4a and 4b?

2.12 We will address these questions with a more complete version of the legends for figures 4a and 4b in the revised manuscript.

The legendof this figure was completed and improved

- …Provide a more nuanced discussion of mismatch between bulk concentrations displayed in Fig 4a.

2.13 The chemistry derived from previously published data comes from:

1) Lake sediment samples in which the glass content was separated from the sediment matrix
2) (Bulk) pumice fragments collected from outcrops of tephra deposits.

In both mentioned cases, the samples were enriched in pyroclastics previous to measurements. For samples presented in this study, pyroclastic fragments and sediment were not separated. Therefore, it is likely the SiO2 concentration within our samples (the lowermost limit of T5) is hindered by being mixed with minerogenic clasts or pyroclastic crystals present in the lake sediments. We will extend our explanation of this matter for the revision.

We provided more details on discrepancies between our bulk samples and previous publications

- Also, restructuring is needed here, as the age model is not presented yet and information on the activity of the two volcanoes is too late in this chapter. Start directly with activity of volcanoes in the region to confine the origin of T5.

2.14 In lines 331-335 within the subchapter "4.5 Chronology" the age assigned to tephra T5 (previous subchapter 4.2, "Tephra records and tephrochronology") is mentioned. Both chapters are directly related. Therefore, the "correct" relative order between them it is arguable.

We accept to reassess which would be the best order to expose our chronology throughout the manuscript.

The order between this 2 subchapters was not changed, as we consider that with the more detailed text that we provided the addressed issues should be solved.

- In chapter 4.3 trends of the magnetic susceptibility are described. However, it seems that these results have no further implications. So why are the data shown and described? Either include the results in the discussion or delete mag sus from the entire text, if it is useless.

2.15 The magnetic susceptibility values and its correlation are used to sustain the association between the element Ti and the magnetic minerals described to be present in the matrix of the basaltic units of the Meseta (lines 358-361, Espinoza et al., 2005).

No changes required

- Chapter 4.5 describes the radiocarbon ages and the establishment of the age-depth model. This is the second major weakness of the manuscript. Although the authors write that "all ages were considered for calculation of the age/depth model" and "T5 ... is used as a temporal control point", it seems that all 14 C ages were regarded as being reliable, except of the age at 240.5 cm composite depth. Moreover, although the authors declare that the age of the potential H1 tephra was used only as a control point, it seems that the age range of the tephra was included in the modelling approach, as the error bar (greyish boundaries in Fig. 6) is wide at the top of the unit F and is not restricted to the age and error bars of sample D-AMS 043189, as this is the case in all other dated horizons.

2.16 As explained in the manuscript, all radiocarbon ages including the one at 240.5 were considered in the age/depth model. The widening of the shaded area (which indicates a 95% confidence of the model) is actually a consequence of the time range between samples at 198 (D-AMs 043189) and 240.5 cm (D-AMS 043190) and the smoothness of the model. Their age difference is large relative to their depth difference (the 30 cm of T5 and unit F were excluded from age-depth calculations). For these reasons the model widens by including both ages with their 95% confidence intervals.

The subchapter on which our chronology is described and discussed was improved in order to clarify misunderstandings

- Table 2 does not contain calibrated mean ages, it shows rbacon derived ages that apparently including age information of the T5 tephra. This would explain the discrepancy between the calibrated mean age of sample D-AMS 043190 in Table 2 compared to the location of the respective sample in Fig. 6. If the tephra was included as a single event, ages of the tephra should be completely removed from the rbacon calculation.

2.17 There is indeed a mixed up with the values in table 2. We will update this table and provide calibrated mean ages. We appreciate pointing out this error.

The table was corrected

- Although I can partly follow the discussion of the impact of the tephra and potential fall-out deposits on top of T5, I miss a thorough discussion, why the age at 240.5 cm depth is somewhat erroneous in contrast to all other radiocarbon ages.

2.18 The 240.5 age is not erroneous, it just deviates from the calculated mean of the age model, but is still included. It is explained in the manuscript that this could be related to our modeling for both the T5 and Unit F (reworked fine ash) as one instantaneous event. It is not possible to have certainty of whether the reworked unit F was event-like in comparison with sedimentation in the lake or deposited in a certain (unknown) amount of time. If the latter is true, then the slump section of the sediments would be shorter and this would allow a smoother transition from the age at 198 to the age at 240.5 cm (composite depth). Additionally, the upper limit of unit F is not clearly defined and thus could also be considered as thinner.

See response 2.16

- Including sample D-AMS 043190 as a reliable sample would lead to a basal age of T5 of ca. 9400 cal yr BP, which is distinctly older than the published ages of the H1 eruption.

2.19 Age 240.5 is part of the model, from which the mean age of 8,277 cal yr BP (age range 8,987 and 8,018 cal yr BP) was derived for the T5-F Unit section. This age is a result of the T5-Unit F section being more influenced by the ~8,020 cal yr BP date positioned immediately above Unit F rather than by the age of ~9,535 cal yr BP located ~10 cm below T5. Even if the age-depth model was more tilted towards the 240.5 cm (composite depth) sample, it would provide a basal age of ~8,600 cal yr BP for H1. This date would still be within a reasonable age range compared to published ages for H1 (mean ages range between 8,585-8,200 cal yr BP for lake cores, Stern et al., 2016).

We do not completely understand on which basis RC2 obtained an age of 9,400 cal yr BP for T5, since its basal boundary is 10 cm above ~9,535 cal yr BP age. Nonetheless, we agree on giving a better explanation for our age/depth model.

See response 2.16

- This provokes a more nuanced discussion of the tephrostratigraphical correlation. On the other hand, assuming that the 14.5 cm on top of T5 may represent a mixture of erosion of sediments and tephra from the catchment is ok, but also this might contain some time. Extrapolating the sedimentation rate calculated from the two ages on top of the tephra downward and including these 14.5 cm of unit F would provide an age slightly older than that of sample D-AMS 043189 i.e. > 8100 cal yr BP (also the 8257 cal yr BP in Table 2 seem to be biased by including a H1 age into the rbacon modelling). Such an age would at least marginally match with published ages of the H1 tephra.

2.20 Yes, this possibility is mentioned. It is one of the unsolvable uncertainties for age-depth modelling that includes reworked pyroclastic material. We tried age-depth modelling with the entire Unit F as a product of 'normal sedimentation,' yielding a period as long as ~650 years of accumulated reworked tephra (and an estimated age of H1 ~8,840 cal yr BP). This result, however, is also arguable. Additionally, the lack of a well-defined upper limit for Unit F introduces further uncertainties to the chronology.

Regardless of this discussion, as explained by RC2, the correlation between T5 and H1 works for our preferred age-depth model as well as for the one with Unit F having time relevance.

See response 2.16

- Overall, a substantial revision is needed in this chapter to provide reliable ages and discuss potential errors in the age-depth modelling. Simply stating that one age has a potential offset and this age is not important and therefore was not excluded is not sufficient for a thorough scientific study and a discussion of potential tephra correlation.

2.21 That is not the explanation we gave, as no sample were excluded. We don't believe that a substantial revision is needed. In our opinion this chapter just requires a more detailed explanation.

See response 2.16

- The Discussion (chapter 6) is very sound and covers a broad suite of environmental factors that control sedimentation in the lake basin. With respect to the above-mentioned broadened discussion of the age model and the reliability of sample D-AMS 043190, the discussion of the Sedimentary environments (chapter 6.2) could be complemented by a discussion of a potential glacial advance in the early Holocene and coarser sedimentation in unit G, probably corresponding with other observations in the broader vicinity (e.g. Douglass et al 2005a and other references therein; glacial advances around 9.4 cal ka BP).

2.22 As mentioned in the manuscript, in lines 292-294 "In the case of Unit G, the slightly higher values for Ca, Ti and K are most likely associated to a high content of crystalline components accumulated in this unit by reworking of T6. Therefore, this increase is not associated to environmental influences". Based on the X-ray data and what we observed under the microscope, the grainsize peak of Unit G is related to an anomalous concentration of fragmented crystalline material, which shows evidence of having a pyroclastic origin. These crystals are well distributed within the sediment matrix, and therefore we included the whole of Unit G within the age-depth model. Consequently, we do not relate this unit to environmental changes, such us allochthonous input produced by an episode due to a glacier advance or higher precipitation.

We provided further details on Unit G in the MS

It is possible that Early Holocene glacier advances occurred in the Meseta Chile Chico and in other locations of Central West Patagonia. However, we consider that our results do not contribute to this discussion, since we do not have a distinctive environmental signature before 7,000 cal yr BP. Moreover, the occurrence of an Early Holocene glacier advance based on cosmogenic dating of Fachinal Moraines remains unclear, as the ages of 8.5 and 6.5 ka from Douglass et al. (2005) present large uncertainties: Lines 432-434 "there is large scatter as ages from two neighboring recessional moraines, i.e. 20.3-9.4 ka for the older moraine and 11-5.8 ka for the younger moraine, were treated individually rather than using a calculated mean (Davies et al., 2020)."

**Specific comments:**

- Line 305: I would not say that TN concentrations of <1.7 % are negligible. A high number of lakes has TN concentrations of <1%, so concentrations of up to 1.7 % are substantial. The high concentrations confirm that OM is an important sediment component.

2.23 Nitrogen concentrations are not used for interpretations, since we always consider total organic matter percentages/Xrf incoherent-coherent values. We will remove this proxy from the revised manuscript.

We deleted Nitrogen and Sulfur

- Lines 393–400: Why is Unit G in stage 1 ignored? It stands out for its high clastic sediment input. Include G in the description and discuss, what may have caused this unit. Please check also for the age of unit G with respect to a revision of the age-depth model. Considering sample

D-AMS 043190 as a reliable sample would place unit G into a period around 9400 cal yr BP or so.

This was answered in the previous section.

We provided further details on Unit G in the MS

Figure 1: Check coordinates in Figure 1b. West and South axes are mixed up.
Figure 3: What is the black triangle in T5 indicating? Include in legend. Litho-boundaries seem not match exactly with the boundaries mentioned in the text.
Table 2. An overview of how the individual core segments were spliced to a core composite would help to better evaluate the reliability of the age-depth model.
Line 58/59: ".... and form part of its drainage basin that currently drains into Río Jeinemeni." check sentence
Line 60: What do you mean with "topographic connections" of the glacial cirques?
Lines 83: Include coordinates in Fig. 2a, provide a scale bar in 2b indicating the dimensions of the Laguna Meseta.
Lines 84/85 and also throughout the entire text: Check Fig 1.c or Fig. 1.c etc for correct spelling
Line 140: ...were performed, a 3-m-long ....
Line 146/147: "Due to their compositional variability, no element ratio correlates with all six tephra layers. Therefore, data points corresponding to these layers were discarded for
correlation coe#cient calculations." – I assume that the element ratios are based on the XRF scanner data? I do not understand these sentences or the relation to what is written in the XRF scanning chapter. Can you rephrase?
Line 236: delete "sediment"
Line 238: Plant remains are not shown in Figure 3
Lines 244-245: What do you mean with grainsize is very well sorted? I assume that this is true for individual horizons in Unit B, but not for the entire Unit. Pls specify.
Line 247: with plant remains of up to a few millimeters in size
Line 248: Chapter header should be "Tephra layers"
Line 250: XRF scanning results show a heterogenous composition of these layers, re%ecting di$erent volcanic sources
Line 261: I have not seen the age model yet. Moreover, I have no information on the activity of Mentolat and Hudson yet. This comes later in the text. Restructuring needed.
Line 289: The highest values of Ca are in unit B, not in T5, please correct, check also for other elements. Rewording needed.
Line 296: " ... is obscured by T3 and T4, values for chemical elements..."

All remaining specific comments correspond to formatting suggestions, and these will be taken into consideration during the next stage of the manuscript.

All comments were assessed and considered

**References:**

Davies, B. J., Darvill, C. M., Lovell, H., Bendle, J. M., Dowdeswell, J. A., Fabel, D., García, J. L., Geiger, A., Glasser, N. F., Gheorghiu, D. M., Harrison, S., Hein, A. S., Kaplan, M. R., Martin, J. R. V., Mendelova, M., Palmer, A., Pelto, M., Rodés, Á., Sagredo, E. A., Smedley, R. K., Smellie, J. L., and Thorndycraft, V. R.: The evolution of the Patagonian Ice Sheet from 35 ka to the present day (PATICE), Earth Sci. Rev., 204, 103152, https://doi.org/10.1016/j.earscirev.2020.103152, 2020.
Douglass, D., Singer, B., Kaplan, M., Ackert, R., Mickelson, D., and Caffee, M.: Evidence of early Holocene glacial advances in southern South America from cosmogenic surface-exposure dating, Geology, 33, 237-240, 10.1130/G21144.1, 2005.

Espinoza, F., Morata, D., Pelleter, E., Maury, R., Guivel, C., Suárez, M., Lagabrielle, Y., Polve, M., Bellon, H., Cotten, J., and R, D.: Petrogenesis of the Eocene and Mio-Pliocene alkaline basaltic magmatism in Meseta Chile chico, southern Patagonia, Chile : evidence of two slab window processes, Lithos, 3-4, 314-343, 2005.

Kilian, R. and Lamy, F.: A review of Glacial and Holocene paleoclimate records from southernmost Patagonia (49–55°S), Quat. Sci. Rev., 53, 1-23, https://doi.org/10.1016/j.quascirev.2012.07.017, 2012.

Moreno, P. I., Vilanova, I., Villa-Martínez, R., Dunbar, R. B., Mucciarone, D. A., Kaplan, M. R., Garreaud, R. D., Rojas, M., Moy, C. M., De Pol-Holz, R., and Lambert, F.: Onset and Evolution of Southern Annular Mode-Like Changes at Centennial Timescale, Sci. Rep., 8, 3458, 10.1038/s41598-018-21836-6, 2018.

Naranjo, J. A. and Stern, C. R.: Holocene tephrochronology of the southernmost part (42°30'-45°S) of the Andean Southern Volcanic Zone, Rev. Geol. Chile, 31, 224-240, 2004.

Weller, D., De Porras, M., Maldonado, A., Méndez, C., and Stern, C.: Holocene tephrochronology of the lower Río Cisnes valley, southern Chile, Andean Geol., 44, 229-248, 10.5027/andgeoV44n3-a01, 2017.

Zolitschka, B., Anselmetti, F., Ariztegui, D., Corbella, H., Francus, P., Lücke, A., Maidana, N. I., Ohlendorf, C., Schäbitz, F., and Wastegård, S.: Environment and climate of the last 51,000 years – new insights from the Potrok Aike maar lake Sediment Archive Drilling prOject (PASADO), Quat. Sci. Rev., 71, 1-12, https://doi.org/10.1016/j.quascirev.2012.11.024, 2013.